# 14-3-3 protein augments the protein stability of phosphorylated spastin and promotes the recovery of spinal cord injury through its agonist intervention

Qiuling Liu[†], Hua Yang[†], Jianxian Luo[†], Cheng Peng, Ke Wang, Guowei Zhang, Hongsheng Lin*, Zhisheng Ji*

Department of Orthopedics, The First Affiliated Hospital of Jinan University, Guangzhou, China

*For correspondence:
tlinhsh@jnu.edu.cn (HL);
tzhishengji@jnu.edu.cn (ZJ)

[†]These authors contributed equally to this work

Competing interest: The authors declare that no competing interests exist.

**Abstract** Axon regeneration is abortive in the central nervous system following injury. Orchestrating microtubule dynamics has emerged as a promising approach to improve axonal regeneration. The microtubule severing enzyme spastin is essential for axonal development and regeneration through remodeling of microtubule arrangement. To date, however, little is known regarding the mechanisms underlying spastin action in neural regeneration after spinal cord injury. Here, we use glutathione transferase pulldown and immunoprecipitation assays to demonstrate that 14-3-3 interacts with spastin, both in vivo and in vitro, via spastin Ser233 phosphorylation. Moreover, we show that 14-3-3 protects spastin from degradation by inhibiting the ubiquitination pathway and upregulates the spastin-dependent severing ability. Furthermore, the 14-3-3 agonist Fusicoccin (FC-A) promotes neurite outgrowth and regeneration in vitro which needs spastin activation. Western blot and immunofluorescence results revealed that 14-3-3 protein is upregulated in the neuronal compartment after spinal cord injury in vivo. In addition, administration of FC-A not only promotes locomotor recovery, but also nerve regeneration following spinal cord injury in both contusion and lateral hemisection models; however, the application of spastin inhibitor spastazoline successfully reverses these phenomena. Taken together, these results indicate that 14-3-3 is a molecular switch that regulates spastin protein levels, and the small molecule 14-3-3 agonist FC-A effectively mediates the recovery of spinal cord injury in mice which requires spastin participation.

## eLife assessment

The finding that Fusicoccin (FC-A) promotes locomotor recovery after spinal cord injury is **useful** and is supported by **solid** data, and the idea of harnessing small molecules that may affect protein-protein interactions to promote axon regeneration is **valuable**. The evidence showing that 14-3-3 and spastin interact and that 14-3-3 enhances spastin function and stability in cells is **solid**.

## Introduction

Regeneration of the adult central nervous system neurons is difficult after injury, a phenomenon that results in permanent neurological impairments (*Curcio and Bradke, 2018*). Researchers have attributed this not only to the highly inhibitory extrinsic environment at the injury site but also to limited cell-intrinsic growth capacity. Many studies have demonstrated that re-programming of growth-associated processes plays a key role in manipulation of axonal regeneration (*Hilton et al., 2022*; *Mahar and Cavalli, 2018*; *Clark et al., 2014*). However, the specific mechanisms underlying

axonal regeneration failure remain largely unknown. Cytoskeleton remodeling, including actin and microtubule (MT) reorganization, appears to be the key factors directing the fate of axonal regeneration and sprouting (*Jin, 2015*). Notably, the retraction bulb of severed axons at the lesion site of the spinal cord needs a dynamic cytoskeleton for successful regeneration. Moreover, the formation of axon collaterals from spared axons also needs dynamic cytoskeleton reorganization to form new neural circuits with the innervated neurons (*Pinto-Costa et al., 2020*; *Crunkhorn, 2015*). Mechanistically, invasion by MTs can not only provide mechanical force, but also guide intracellular transport to regulate distribution of specific molecules, including mitochondria, peroxisomes, growth factors and others, and subsequently mediate axonal extension. Therefore, timely invasion of microtubules can not only power the formation of a growth cone with competent growth ability from the retraction bulb, but also improve the process of axonal sprouting from the spared axons (*Chen and Rolls, 2012*; *Vargas et al., 2020*). Overall, manipulating MT dynamics is critical for boosting axon regeneration.

Spastin is encoded by *SPAST* gene and its mutations was identified in patients with hereditary spastic paraplegia (HSP) (*Shribman et al., 2019*). It is an MT severing enzyme (including spastin, katanin, and fidgetin) which can cut long MTs into short fragments, a phenomenon that not only endows MTs with high dynamics but also mediates their subsequent turning and regrowth under physiological conditions (*Liu et al., 2021*). Spastin has two major isoforms, namely M1 and M87, coded form different translation initiation codons (1$^{st}$ and 87$^{th}$ methionine). Both isoforms are highly expressed in the spinal cord (*Salinas et al., 2007*). Moreover, *SPAST* mutation can lead to the degeneration of the corticospinal tracts in HSP patients. In addition, spastin is highly expressed in the central neuronal system (CNS) and essential for formation of axon branches as well as their extension during neural development (*Lopes et al., 2020*; *Ji et al., 2020*). These evidences indicate that spastin plays an important role in axonal development and maintenance in the spinal cord. Intriguingly, a study conducted in *Drosophila* has reported that excess or deficient dosage of spastin is detrimental to axon regeneration (*Stone et al., 2012*). Furthermore, another study in *Drosophila* has also indicated the significance of spastin in coordinating the concentration of the endoplasmic reticulum (ER) and microtubules at the growing axon tips during axon regeneration (*Rao et al., 2016*). In a separate study, researchers administered spastin conjugated with polyethylene glycol into the epineurium and observed its potential in mediating the recovery of motor function following sciatic nerve injury. However, this study lacks the requisite histological and molecular biological evidence (*Lin et al., 2019*). Currently, there is insufficient research to definitively establish the role of spastin in axon regeneration after central nervous system injury in adult mammals. Manipulating spastin expression seems to be an effective way to promote axon regeneration. However, efficient regeneration is dependent on the precise spastin dosage, with studies showing that excess spastin is toxic and may destroy the MT network, while low doses have been associated with insufficient MT severing which subsequently limits MT remodeling for regeneration (*Stone et al., 2012*; *Rao et al., 2016*). Therefore, direct manipulation of the *SPAST* gene may not be the most appropriate approach for promoting successful axon regeneration after the CNS injury. It is worth noting that post-translational modifications, protein metabolism, and protein-protein interactions are vital for regulating the protein's activity (*Martins-Marques et al., 2019*). Thus, elucidating the specific molecular mechanism underlying spastin action is imperative to guiding the future development of effective interventions to promote axon regeneration following damage to the CNS.

14-3-3 protein is a type of adaptor protein that plays important roles in many signaling pathways through interaction with their substrates (*Fu et al., 2021b*; *Fu et al., 2021a*). For instance, it inhibits the ROS-induced cell death by interacting with FOXO transcription factors to prevent upregulation of many pro-apoptotic genes (*Nielsen et al., 2008*). In mammals, seven isoforms, namely β,γ,ε, ζ , η ,θ, and σ, have been identified. They are not only highly conserved in the mammalian system, but also form a dimeric structure to interact with their substrates which are mostly phosphoproteins (*Park et al., 2019*). Previous studies have shown that spastin is phosphorylated in Ser233 which showed its potential for interaction with 14-3-3 (*Pisciottani et al., 2019*). In addition, 14-3-3 protein is also highly expressed in CNS where it plays a vital role in both axonal growth and guidance. Previous studies have also demonstrated that 14-3-3 promotes cortical neurite outgrowth and axon regeneration (*Kaplan et al., 2017*; *Yam et al., 2012*). To date, however, the molecular mechanisms underlying its action in the promotion of axon regeneration as well as its role in spinal cord injury remain largely unknown.

In this study, we demonstrate that 14-3-3 interacts with spastin to regulate its stability, thus controlling both neurite outgrowth and regeneration after injury. Our results reveal that 14-3-3 interacts with Ser233 in spastin, in a phosphorylation-dependent manner, thereby preventing its degradation by ubiquitination and upregulates the severing ability via spastin-mediated activity. Moreover, we found that the 14-3-3 agonist FC-A application resulted in enhanced nerve regeneration and locomotor improvement both in the contusion model and in the T10 lateral hemi-transection model of spinal cord injury. Collectively, our findings suggest 14-3-3 and spastin are novel intervention targets for nerve regeneration after spinal cord injury.

## Results

### 14-3-3 protein interacts with spastin

Spastin, an MTs severing enzyme, can cut long MTs into short fragments thereby contributing to axonal elongation and branching. However, excess spastin levels by gene manipulation can not only destroy the MT cytoskeleton but are also toxic to neurons. Unraveling the underlying mechanisms of spastin regulation under physiological conditions may provide novel insights into the mode of axon regeneration following CNS damage. In our previous study (*Ji et al., 2021*), GST-spastin pulldown assays with proteins lysed from the spinal cord tissue were performed, and the obtained proteins were analyzed via mass spectrometry to determine the underlying molecular mechanisms of spastin. The peptides that interacted with GST were subtracted from those interacting with GST-spastin, and the peptides of proteins that interacts with spastin were obtained. We revealed 14-3-3 as putative spastin-interacting proteins (*Figure 1A*, *Figure 1—figure supplement 1*). Notably, 14-3-3 proteins consist of seven isoforms that are highly conserved and act by targeting phosphoserine and phosphothreonine motifs of substrate proteins. We have detected many different 14-3-3 peptides in the complex, including LAEQAER, NLLSVAYK, and AVTEQGAELSNEER (*Figure 1—figure supplement 1A*). Notably, these peptides were highly conserved among different isoforms (*Figure 1—figure supplement 1B*). Next, we employed immunoprecipitation to determine the biochemical interaction between 14-3-3 protein and spastin, and found that they endogenously interacted in the spinal cord tissue and the cortical neurons (*Figure 1B and C*). To elucidate the distribution of 14-3-3 and spastin within neurons, hippocampal neurons were double stained with spastin and 14-3-3 antibodies. The results revealed that 14-3-3 exhibited a characteristic distribution in axons, including aggregation at the growth cone and specific locations in the axon shaft, while spastin also showed accumulations at the sites where 14-3-3 proteins aggregated. Colocalization profiles of the dotted box are displayed in *Figure 1E*. 14-3-3 proteins consist of seven isoforms, and their protein spatial structure are highly conserved (*Figure 1F*). To further identify which isoform of 14-3-3 interacts with spastin, genes of six 14-3-3 isoforms were obtained from rat brain cDNA and inserted these fragments into the pEGX-5X-3 vector. Subsequently, GST 14-3-3 fusion proteins were expressed and purified in vitro (*Figure 1G*). Results from a GST pull-down assay revealed that all 14-3-3 isoforms could interact with spastin (*Figure 1H*). Subsequent co-immunoprecipitation assay results also confirmed that all 14-3-3 isoforms could form complexes with spastin (*Figure 1I*). Collectively, these results indicated that the 14-3-3 protein formed a protein complex with spastin both in vitro and in vivo.

### 14-3-3 binds to phosphorylated Ser233 in spastin

Next, we characterized the interaction region of spastin/14-3-3 binding. To this end, we generated several GFP-tagged deletion constructs based on the modular domain of spastin (*Figure 2A*), then co-expressed them with Flag-tagged 14-3-3 into HEK293T cells. Co-immunoprecipitation assay, using GFP-Trap beads, revealed that spastin fragments containing amino acids 85–316 but not those with 85–214 coprecipitated with 14-3-3 (*Figure 2B*). Next, we performed immunoprecipitation assays using the region (215–336 amino acids) of spastin with 14-3-3 and found that 14-3-3 specifically binds to the region (215–336 amino acids) of spastin (*Figure 2C*).

Previous studies have shown that phosphorylation of the 14-3-3 substrate regulates the binding of 14-3-3 proteins, although 14-3-3 can also bind to non-phosphorylating proteins. To investigate whether spastin's phosphorylation state affects its binding with 14-3-3, we performed pulldown assays by using GST-14-3-3 and GFP-spastin from transfected HEK293T cell extracts in the presence of broad protein kinase staurosporine. To validate whether staurosporine could decrease the phosphorylation

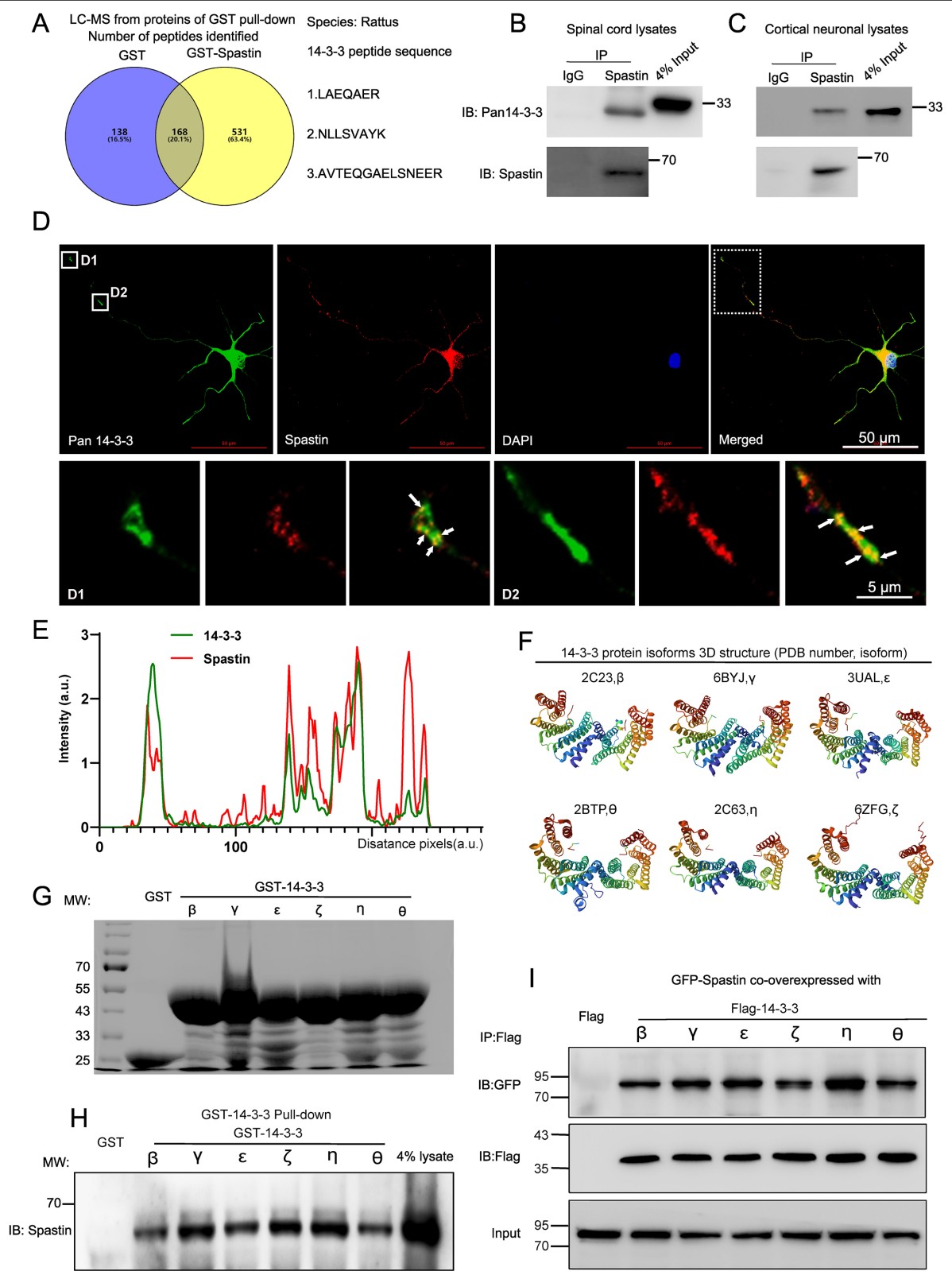

**Figure 1.** 14-3-3 interacts with spastin in vivo and in vitro. (**A**) Proteins from the spinal cord tissue (approximately 1 cm around the T10 level) were pulldowned by purified GST and GST-Spastin proteins and the proteins obtained were subjected to mass spectrometry analysis. The 306 peptides were identified in the GST group and 699 peptides in the GST-Spastin group. The peptides that were common to both the GST and GST-Spastin groups were excluded, and 531 peptides were the peptides of proteins that interact with spastin. Three 14-3-3 protein peptides were shown. (**B**) Immunoprecipitation

*Figure 1 continued on next page*

*Figure 1 continued*

assay using spastin antibody was performed from spinal cord lysates and Pan 14-3-3 antibody was used for western blotting. (**C**) Immunoprecipitation assay using spastin antibody was performed from cortical neural lysates and Pan 14-3-3 antibody was used for western blot. (**D**) Hippocampal neurons were stained with 14-3-3 (green) and spastin (red) antibodies, the growth cone (D1) and axon shaft (D2) profile were shown. Scale bar: 50 μm, 5 μm. (**E**) Localization of spastin (red line) and 14-3-3 (green line) protein in the hippocampal neurons were analyzed and the intensity of the white dotted line across the neurite compartment were measured. (**F**) The highly conserved spatial conformation of different 14-3-3 protein isoforms were shown. (**G**) Purified GST and GST-14-3-3s fusion proteins were subjected to sodium dodecyl-sulfate polyacrylamide gel electrophoresis and coomassie blue staining was confirmed the successful purification of relative proteins. (**H**) GST fusion proteins were used to pull down the lysates of the spinal cord and spastin antibody was used for western blotting. (**I**) HEK 293T cells were transfected with Flag-tagged 14-3-3s and GFP-tagged spastin, the cell lysates were subjected to GFP-Trap assay, and Flag antibody was used to detect the presence of 14-3-3s.

The online version of this article includes the following source data and figure supplement(s) for figure 1:

**Source data 1.** Pulldowned peptides list from the mass spectrum.

**Source data 2.** Raw and annotated immunoblots for *Figure 1*.

**Source data 3.** Numerical data for *Figure 1E*.

**Figure supplement 1.** The peptides of 14-3-3 essential for the interaction with spastin are highly conserved in all 14-3-3 isoforms.

levels of spastin, we enriched spastin through immunoprecipitation and used Pan phosphoserine/threonine antibody to observe spastin's phosphorylation. The results showed that staurosporine significantly reduces the phosphorylation levels of spastin (*Figure 2D*). Furthermore, our results revealed that the presence of staurosporine markedly weakened the binding between spastin and 14-3-3, indicating that phosphorylation of spastin is sufficient for the binding with 14-3-3 (*Figure 2E*).

Next, we predicted five putative 14-3-3 binding sites in spastin, namely amino acids 213–218, 230–235, 243–249, 425–460, and 559–564, which match the motif of RSX(S/T)XP. Combination with phosphorylation of spastin reported from MS data (https://www.phosphosite.org/proteinAction. action?id=8325&showAllSites=true), we revealed two putative binding motifs (Ser233 and Ser562) in spastin as putative targets for 14-3-3 binding (*Figure 2F*). Widely employed mutation techniques have shown that mutating a protein's phosphorylation site to alanine prevents phosphorylation, while mutating it to aspartic acid can mimic phosphorylation. Then, we mutated single Ser233 and Ser562 or both to alanine in the spastin tagged with GFP and co-expressed them with Flag-tagged 14-3-3 in HEK293T cells. Immunoprecipitation of the lysates, using an antibody against GFP, revealed that spastin S562A can bind with 14-3-3, while spastin S233A and spastin Pan A cannot (*Figure 2G*). To further illustrate the significance of phosphorylation of spastin at S233 in its interaction with 14-3-3, the mutation of Ser233 in spastin to aspartic acid was also transfected for immunoprecipitation, the results revealed that spastin S233D exhibited enhanced binding to 14-3-3 compared to both wild-type and spastin S233A, and this interaction was even insensitive to the staurosporine treatment (*Figure 2H*). This evidence further suggests that phosphorylation of spastin at the S233 site is a critical event in its interaction with 14-3-3. Also, these results were consistent with those in *Figure 3B–C*, where Ser233 was localized in the region of 215–316 amino acids in spastin, further indicating that the binding of 14-3-3/spastin requires phosphorylation of Ser233 in spastin. Meanwhile, upon transfecting GFP-tagged spastin S233A and S233D mutants into COS7 cells, we observed that both spastin S233A and S233D mutants effectively severed microtubules, which indicates that the phosphorylation of the spastin S233 residue does not affect spastin's ability to sever microtubules (*Figure 2—figure supplement 1*).

To examine whether Ser233 in spastin is involved in neurite outgrowth, wild-type (GFP-Spastin) or spastin Ser233 mutant (GFP-Spastin S233A or GFP-Spastin S233D) was moderately expressed in primary hippocampal neurons at 2$^{nd}$ DIV. It is important to note that overexpression of spastin can lead to the severing of the overall microtubule cytoskeleton in neurons. To achieve a moderate level of spastin expression, the transfection dosage and duration were meticulously controlled. Consistent with earlier results (*Jiang et al., 2020*; *Riano et al., 2009*), spastin promoted neurite outgrowth, as evidenced by both the length and branches of neurite. However, the promotion effect of spastin on neurite outgrowth is abolished when Ser233 was mutated to alanine, whereas the promotion effect of spastin on neurite outgrowth is significantly enhanced when Ser233 was mutated to aspartic acid. These findings suggest that the phosphorylation of Ser233 in spastin exerts a positive effect on neurite outgrowth during development.

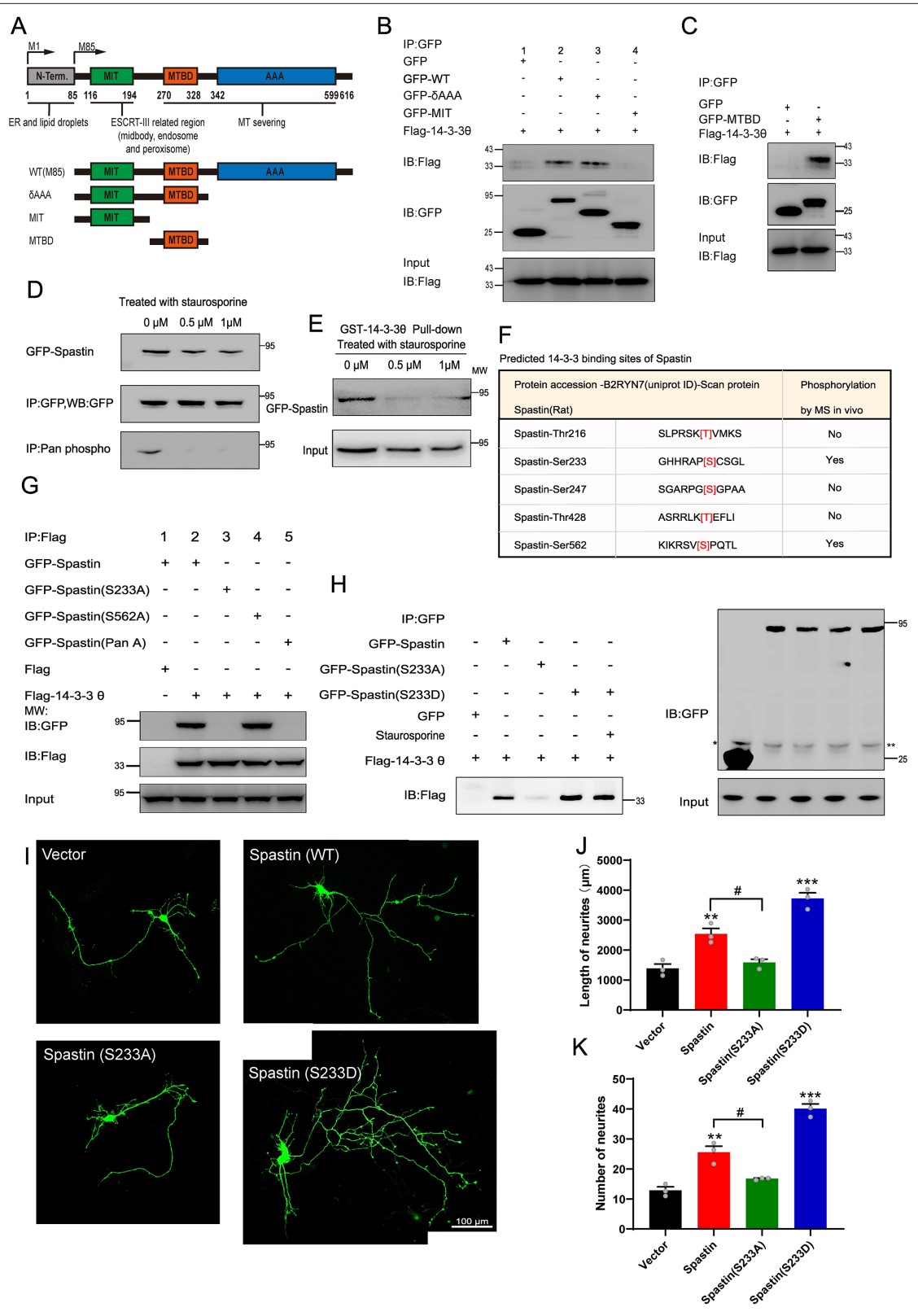

**Figure 2.** Ser233 in spastin is essential for the interaction with 14-3-3 and important for neurite outgrowth. (**A**) Schematic illustration of the gene truncation strategy for the modular domain of spastin. (**B**) Truncation mutants of spastin together with Flag-tagged 14-3-3 were transfected into HEK 293T cells, and cell lysates were immunoprecipitated by GFP-Trap assay. Flag antibody were used for detecting the flag-tagged 14-3-3 fusion protein. (**C**) The microtubule-binding domain (MTBD) truncated GFP-tagged spastin were transfected together with Flag-tagged 14-3-3 into HEK 293T cells,

*Figure 2 continued on next page*

*Figure 2 continued*

and subjected to GFP trap assay. (**D**) GFP-tagged spastin was transfected into HEK293T cells by treatment with staurosporine (a broad spectrum protein kinase inhibitor) or not. GFP-Trap beads were used to enrich GFP-Spastin, and the phosphorylation levels of spastin were detected using a Pan phosphoserine/threonine antibody. (**E**) HEK 293T cells were transfected with GFP-spastin and then treated with or without staurosporine for 1 hr. Cell lysates were pull downed by GST 14-3-3 fusion protein and GFP antibodies were used for western blotting. (**F**) The binding sites between 14-3-3 and spastin were predicted using the scansite 2.0 software and whether these sites could be phosphorylated were confirmed by the phosphosite plus website. (**G**) Ser233 and Ser562 of spastin were mutated to alanine, these GFP fusion mutations were transfected together with Flag-tagged 14-3-3, and then cell lysates were subjected to GFP-trap assay. Flag antibody was used to detect the binding ability between spastin mutations and 14-3-3. (**H**) GFP-tagged wild-type spastin, spastin S233A (mimicking dephosphorylation), or spastin S233D (mimicking phosphorylation) were co-transfected with Flag-tagged 14-3-3 into HEK293T cells. The group transfected with GFP-Spastin S233D was additionally treated with staurosporine. Cells were lysed and immunoprecipitation were conducted using GFP-Trap, and western blot analysis was performed using Flag antibodies to investigate the impact of Ser233 phosphorylation in spastin on its interaction with 14-3-3. (**I**) Hippocampal neurons were grown on DIV2 and transfected with spastin Ser233 mutations (Spastin S233A, Spastin S233D), then fixed at DIV 3. Representative images of transfected neurons as indicated were shown. Quantitative analysis of the length of neurites (**J**) and the number of neurites (**K**) were shown. Data were represented as the mean value of three independent experiments (n=3). All error bars are SEM. Differences across multiple groups were determined using One-way analysis of variance (ANOVA), followed by Newman-Keuls post hoc tests for mean separations. *p<0.05. Scale bar: 100 μm.

The online version of this article includes the following source data and figure supplement(s) for figure 2:

**Source data 1.** Raw and annotated blots for *Figure 2*.

**Source data 2.** Numerical data for *Figure 2J–K*.

**Figure supplement 1.** The phosphorylation of Ser233 in spastin does not affect the microtubule severing activity of spastin.

## 14-3-3 protects spastin from degradation via spastin S233 phosphorylation

How does the interaction between 14-3-3 and Ser233 in spastin regulate neurite outgrowth during neural development, and why does phosphorylation of spastin in Ser233 further promotes the neurite outgrowth? Hence, we first investigated whether the phosphorylation of spastin at S233 affects its microtubule-severing capability. As illustrated in *Figure 2—figure supplement 1*, both spastin S233A and S233D could effectively severed microtubules. Furthermore, the S233 site is situated outside of spastin's MTBD domain (amino acids 270–328) and AAA (amino acids 342–599) domains. Consequently, we speculated that phosphorylation of the spastin S233 site does not influence spastin's microtubule-severing ability. Notably, we observed a significantly higher green fluorescence intensity of GFP-spastin (S233D) compared to spastin S233A. Combined with the fact that spastin acts in a dose-dependent manner and phosphorylation of spastin in Ser233 further promote the neurite outgrowth. We, therefore, hypothesized that phosphorylation of Ser233 in spastin may affect the protein levels of spastin. To further test whether 14-3-3 would affect spastin stability, we performed chase-time assays via GFP-spastin overexpression, followed by treatment with cycloheximide (CHX) to inhibit protein synthesis. Results showed that expression of GFP-spastin protein was markedly downregulated at 9 hr after CHX treatment (*Figure 3A*). Next, we co-expressed Flag-14-3-3 with GFP-spastin in the same treatment and found that 14-3-3 could significantly protect spastin from degradation after CHX treatment, even after 12 hr (*Figure 3B*). We also investigated whether spastin S233 phosphorylation by transfecting with GFP-tagged spastin S233A or S233D followed by CHX treatment. Due to the presence of endogenous 14-3-3, we did not overexpress 14-3-3 with S233 mutations that mimicked different phosphorylation states. The results demonstrated that phosphorylation of spastin at S233 (S233D) protected spastin from degradation (*Figure 3C&D&I*). Since 14-3-3 can inhibit spastin protein turnover, does 14-3-3 also increase the protein level of spastin? Therefore, we conducted chase-time assays by overexpressing GFP-spastin without the addition of CHX. The results showed that 14-3-3 could upregulate the protein levels of spastin (*Figure 3E&F&J*). We also observed that spastin S233D significantly increased the protein level compared to the wild-type spastin (*Figure 3G&K*). These findings suggest that 14-3-3 can inhibit protein turnover by binding to phosphorylated spastin and upregulate the protein levels of spastin.

A previous study demonstrated that spastin could be polyubiquitinated (*Sardina et al., 2020*). Therefore, we employed MG132, a proteasome inhibitor, to suppress the ubiquitin-proteasome degradation and assess whether it impacts the degradation pathway of spastin in neuronal compartments. The results revealed that in the presence of MG132, the protein levels of spastin significantly increased. However, in the presence of CHX, the protein levels of spastin decreased, while the inhibition

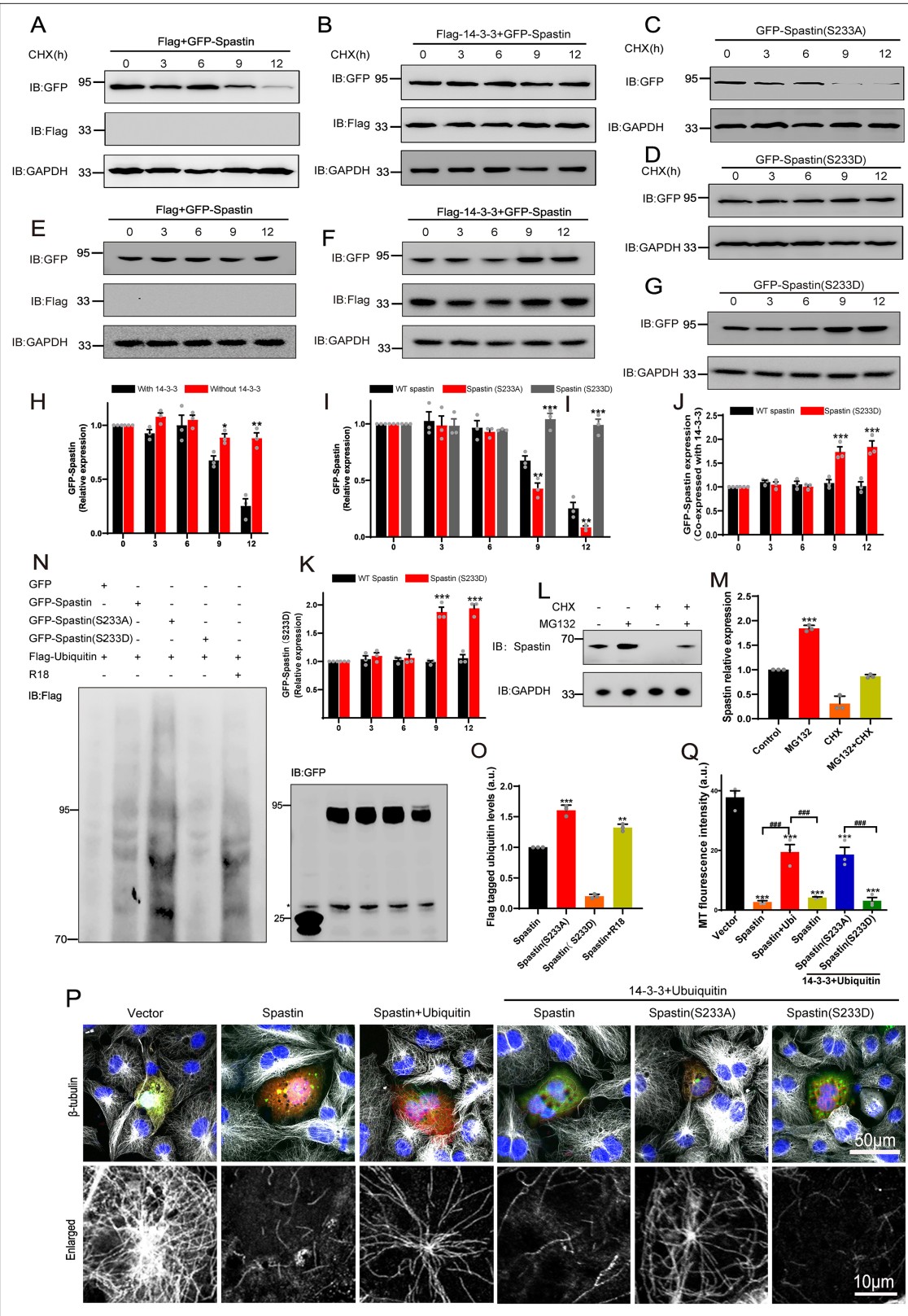

**Figure 3.** 14-3-3 protein protects spastin by Ser233 phosphorylation from degradation through ubiquitin pathway. (**A**, **B**) GFP-tagged spastin together with Flag or with Flag-tagged 14-3-3 was transfected into HEK 293T cells. Following the addition of protein synthesis inhibitor (cycloheximide, CHX), cell lysates were subjected to western blot to detect the GFP-tagged spastin protein levels. (**C**, **D**) Western blot analysis of HEK 293T cell lysates transfected with spastin mutations (S233A or S233D) after treating with cycloheximide at different indicated times. (**E**, **F**) GFP-tagged spastin with

*Figure 3 continued on next page*

*Figure 3 continued*

Flag or Flag-tagged 14-3-3 were co-transfected into HEK293T cells. Cell protein lysates from different time points were collected for western blot. (**G**) GFP-tagged spastin S233D was transfected into HEK293T cells, and western blot experiments were conducted to observe the protein expression levels of GFP-tagged spastin at different time points. (**H**) Quantitative analysis of spastin protein levels after addition of cycloheximide with or without 14-3-3 protein (n=3, *p<0.05). (**I**) Quantitative analysis of GFP-tagged spastin protein levels upon transfected with different spastin mutations (n=3, *p<0.05). (**J**) Quantitative analysis of spastin protein levels with or without 14-3-3 protein (n=3, *p<0.05). (**K**) Quantitative analysis of GFP-Spastin (S233D) protein levels at different time points (n=3, *p<0.05). (**L**) Cultured cortical neurons were treated with proteasome inhibitor MG132 or/and CHX for 36 hr, cell lysates were subjected to western blot analysis. (**M**) Quantification analysis of spastin protein levels (n=3, *p<0.05). (**N**) HEK 293T cells were transfected with GFP-tagged spastin or its S233 mutants and Flag-tagged ubiquitin, together with R18 or not, cells were harvested and subjected to immunoprecipitation using GFP antibody. Flag antibody was used to detect the ubiquitin band. (**O**) Quantitative analysis of the ubiquitination levels of GFP-Spastin (n=3, *p<0.05). (**P**) COS7 cells were transfected with indicated GFP-tagged spastin with or without Flag-tagged ubiquitin for 24 hr. Then, cells were fixed and stained with tubulin antibody to visualize microtubule arrangements. (**Q**) The normalized quantitative analysis of microtubule fluorescence intensity among indicated groups. Data were represented as the mean value of three independent experiments (n=3). All error bars are SEM. *p<0.05, **p<0.01, ***p<0.001. Scale bar: 50 µm, 10 µm.

The online version of this article includes the following source data and figure supplement(s) for figure 3:

**Source data 1.** Raw and annotated blots for *Figure 3*.

**Source data 2.** Numerical data for *Figure 3H*.

**Source data 3.** Numerical data for *Figure 3I*.

**Source data 4.** Numerical data for *Figure 3J*.

**Source data 5.** Numerical data for *Figure 3K*.

**Source data 6.** Numerical data for *Figure 3M*.

**Source data 7.** Numerical data for *Figure 3O*.

**Source data 8.** Numerical data for *Figure 3Q*.

**Figure supplement 1.** 14-3-3 regulates spastin-dependent microtubule severing ability.

of ubiquitin-proteasome degradation by MG132 did not show a significant decrease (*Figure 3L&M*). These findings indicate the presence of a proteasomal degradation pathway for spastin.

Previous studies have shown that FC-A and R18 interact with 14-3-3 protein by either stabilizing or inhibiting the binding of 14-3-3 with their substrates by directly docking to the groove in the 14-3-3 proteins with high efficiency (*Kaplan et al., 2017*; *Petosa et al., 1998*; *Pisa et al., 2019a*). The structure of FC-A and R18 binding with 14-3-3 are shown in *Figure 4A*. To test our hypothesis, we first explored whether FC-A or R18 could enhance or inhibit the interaction between 14-3-3 and spastin by co-immunoprecipitation with lysates of HEK293T cells, which had been transfected with GFP-spastin and Flag-14-3-3 plasmids whether together with FC-A application or R18 transfection. We found that FC-A could enhance the binding of 14-3-3/spastin, as evidenced by more intense bands compared to no FC-A application, whereas R18 could efficiently inhibit the binding of 14-3-3/spastin with almost no binding was detected (*Figure 4B*). To further determine whether 14-3-3 affected spastin's ubiquitination pathway, we performed co-immunoprecipitation assays. GFP-tagged spastin, GFP-tagged spastin S233A, GFP-tagged spastin S233D, and Flag-tagged ubiquitin were transfected into HEK293T cells, together with 14-3-3 binding inhibitor R18 or not. Results indicated that GFP-spastin could be ubiquitinated, while the ubiquitination of non-phosphorylated spastin (S233A) was significantly enhanced, and the addition of the 14-3-3 inhibitor R18 elevated the ubiquitination levels of spastin (*Figure 3N&O*). These results indicated that 14-3-3 protects spastin from degradation by inhibiting the ubiquitination pathway.

To assess the impact of 14-3-3 on the microtubule-severing ability of spastin, we co-transfected GFP-tagged spastin into COS7 cells with or without ubiquitin. COS7 cells are very flat, and hence provide more spatial resolution compared to neurons. Results showed that cells transfected with wild-type spastin significantly induced microtubule severing, as evidenced by almost no MT staining (*Figure 3P*), while ubiquitin inhibited severing, leading to increased microtubule staining. Based on these conditions, further transfection with 14-3-3 plasmid induced less microtubule staining, indicating that 14-3-3 upregulates microtubule severing and the process is mediated by spastin. Furthermore, we investigated whether this effect was dependent on spastin S233 phosphorylation by replacing GFP-spastin with GFP-spastin (S233A or S233D) during transfection. Results showed that 14-3-3 could upregulate the severing ability of spastin S233D, but not S233A (*Figure 3Q*, *Figure 3—figure*

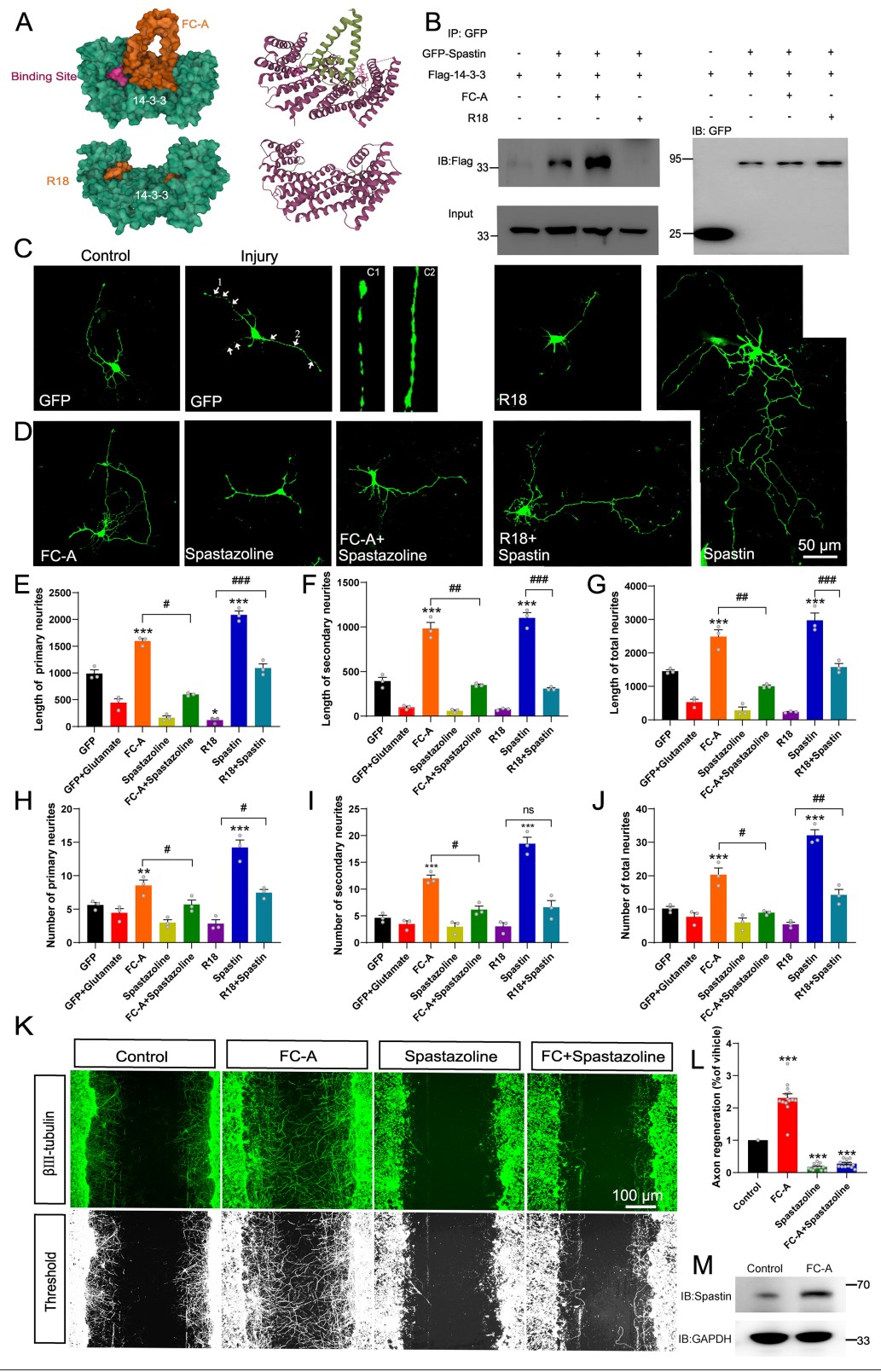

**Figure 4.** 14-3-3 agonist Fusicoccin (FC-A) promotes the repair of neurite outgrowth and regeneration after injury via spastin. (**A**) The surface structure and secondary structure of the molecular docking model between 14-3-3 protein with FC-A (PDB:2O98) or R18 (PDB:1A38). (**B**) HEK 293T cells were transfected with GFP-tagged spastin and Flag-tagged 14-3-3, cell lysates were incubated with FC-A or R18 and then subjected to GFP-trap assay. Flag

*Figure 4 continued on next page*

*Figure 4 continued*

antibodies were used to detect the 14-3-3 binding. (**C**) Hippocampal neurons (DIV2) were transfected with GFP to visualize its morphology. Neurons were then incubated with glutamate (120 µM) to induce injury, the neurite breakages and swellings were found in C1 and C2. (**D**) The injured neurons were transfected with spastin and incubated with FC-A, R18, or spastazoline for 24 hr, then cells were fixed and images were taken by confocal microscopy. Neurons were traced using Image J Pro Plus and quantitative analysis of the length of primary neurites (**E**), secondary neurites, (**F**) and total neurites (**G**) were performed. Quantitative analysis of the number of primary neurites (**H**), secondary neurites (**I**) and total neurites (**J**) from three independent experiments (n=3). *p<0.05, **p<0.01, ***p<0.001. Scale bar: 50 µm. (**K**) Primary cortical neurons were grown and scratch assays were performed at DIV7. FC-A or spastazoline were applied and incubated for 36 hr. Then, cells were stained with βIII-tubulin to visualize the morphology of neurites. (**L**) Cortical neurons cultured in vitro were treated with FC-A for 36 hr, and western blot experiments were conducted to observe the protein levels of spastin. (**M**) Quantitative analysis of the normalized axon regeneration rate. n=14 scratches from three experiments. All error bars are SEM. *p<0.05, **p<0.01, ***p<0.001. Scale bar: 100 µm.

The online version of this article includes the following source data and figure supplement(s) for figure 4:

**Source data 1.** Raw and annotated blots for *Figure 4*.

**Source data 2.** Numerical data for *Figure 4E–J*.

**Source data 3.** Numerical data for *Figure 4L*.

**Figure supplement 1.** 14-3-3 agonist Fusicoccin-A (FC-A) promoted neurite outgrowth of hippocampal neurons in stages 2–3 via spastin.

**Figure supplement 2.** 14-3-3 inhibitor R18 inhibited neurite outgrowth of hippocampal neurons in stages 2–3 via spastin.

**Figure supplement 3.** 14-3-3 agonist Fusicoccin-A (FC-A) promoted neurite regeneration of cortical neurons in via spastin.

---

*supplement 1*), further indicating that 14-3-3 binds to the phosphorylation of S233 in spastin and promotes its microtubule severing ability. Collectively, these results suggest that 14-3-3 protein protects spastin from degradation via spastin S233 phosphorylation.

## 14-3-3 and spastin regulates neurite outgrowth and regeneration after injury

Next, we investigated how the interaction between 14-3-3 and spastin regulates neurite outgrowth and regeneration by applying FC-A and R18 to either stabilize or inhibit 14-3-3 binding. Another drug, spastazoline, introduces an electropositive group to the N386 in the human spastin side chain, leading to specific and efficient inhibition of the ATPase activity of the AAA domain to abolish spastin's microtubule-severing function (*Pisa et al., 2019b*; *Verma et al., 2022*). Subsequently, we used FC-A or R18 together with spastazoline to examine the effects of 14-3-3/spastin protein complex on neurite outgrowth and regeneration.

Since 14-3-3 proteins play important roles in the inhibition of ROS-induced cell death, we first explored the effect of 14-3-3/spastin binding in neurite outgrowth and repair under glutamate circumstances. Following neural injury, neurons undergo lysis, resulting in the release of a substantial quantity of excitatory glutamate (*Savolainen et al., 1998*). Glutamate, upon binding to its receptors, initiates calcium overload, subsequently causing mitochondrial membrane potential collapse, nuclear swelling, and ultimately inducing neuronal necrosis. This process is characterized by neurite swelling and breakage (*Ankarcrona et al., 1995*; *Kajiwara et al., 2009*). Our results showed that the injured neurons had numerous swellings and breakages on the neurite, indicative of successful establishment of an injury model (*Figure 4C*). After an injury, FC-A was administered into primary hippocampal neurons during the developmental stage 2, in combination with application of spastin inhibitor spastazoline. Results demonstrated that FC-A could promote neurite outgrowth, both with regards to lengths and branches, after glutamate-induced injury, while this promoting effect was abolished following spastazoline administration (*Figure 4D–G*). A similar effect was observed in neurons without glutamate-induced injury. In contrast, transfection of R18 resulted in an inhibitory effect on neurite outgrowth, but this was rescued with moderate spastin expression after injury (*Figure 4D–J*). A similar trend was observed in neurons without glutamate-induced injury (*Figure 4—figure supplements 1*

*and 2*). Collectively, these results suggest that 14-3-3 protein promotes the repair of injured neurons through the interaction with spastin.

Next, we investigated the effect of enhanced 14-3-3/spastin binding on neurite regeneration. Briefly, cortical neurons were grown at the 7th DIV and then subjected to scratch assays by drawing a line over the coverslips using a pipet tip. Neurons were then treated with either FC-A or spastazoline for 36 hr. Results showed that FC-A markedly improved neurite regeneration consistent with previous study (*Kaplan et al., 2017*), while spastazoline application almost abolished the enhanced effects of regeneration (*Figure 4K–L*). To further quantify the extend of axon regeneration, we treated cortical neurons with either FC-A or spastazoline for 36 hr, then quantified the longest regenerative axons. Results revealed similar effects (*Figure 4—figure supplement 3*). To further elucidate whether FC-A affects neuronal process by impacting on spastin, cortical neurons cultured in vitro were administrated with FC-A, and the protein expression levels of spastin were detected. The results indicated that FC-A could enhance the expression levels of spastin within cortical neurons (*Figure 4M*). Taken together, these results indicate that spastin-dependent function is a prerequisite for the enhancement of neurite regeneration mediated by FC-A.

## FC-A-mediated nerve regeneration after SCI requires spastin activation

Given that 14-3-3 plays a vital role in neurite outgrowth and axonal regeneration (*Zerr, 2022*), we then hypothesized that 14-3-3 could be involved in the recovery of spinal cord injury. To test this hypothesis, we established a spinal contusion injury model and then employed an immunostaining assay to quantify expression levels of 14-3-3 protein following spinal cord injury. Given the scarcity of neural compartment at the injury center, we select tissue slices as close as possible to the lesion core to illustrate the relationship between 14-3-3 and the injured neurons. The results indicated that 14-3-3 protein was upregulated and the upregulated compartments were principally colocalized in the neuronal compartment (βIII-tubulin labeled) (*Figure 5A*). Western blot experiments also reported that 14-3-3 was upregulated at 3th and 7th days post-injury (DPI), but downregulated at 14 DPI. The 14-3-3 protein level then reached a peak level at 30 DPI (*Figure 5B&E*). Moreover, previous studies have demonstrated that neurons were surrounded by an inhibitory microenvironment 2 weeks after spinal cord injury, our results also showed a slight decrease at 14 DPI, indicating that 14-3-3 acts as an intrinsic switch for neural repair. Moreover, 14-3-3 was reported to be upregulated and acted as a biomarker for diagnosis of Creutzfeldt-Jakob disease (*van den Brand et al., 2012*). Our results revealed that levels of 14-3-3 protein remained high even at 30 DPI, indicating that 14-3-3 may play an important role in the process of recovery after spinal cord injury.

So, how does the expression of spastin change following spinal cord injury? As previously mentioned, the drug FC-A is known to enhance the binding affinity between 14-3-3 and spastin, thereby increasing spastin's protein stability. Therefore, can the administration of FC-A after spinal cord injury upregulate the protein levels of spastin? To address these questions, tissues were collected and western blot experiments were conducted to assess the expression levels of spastin after spinal cord injury. Our findings revealed that the expression of spastin exhibited a sharp decrease immediately after spinal cord injury (1 day), followed by a gradual increase during the acute phase (1–7 days), eventually returning to levels similar to the sham group at 30 days (*Figure 5C*). Notably, the expression trend of spastin is similar to that of 14-3-3 protein after spinal cord injury (*Figure 5H*), suggesting potential involvement of 14-3-3 and spastin in the same pathway. Moreover, we observed that the application of FC-A after spinal cord injury significantly augmented spastin's expression (*Figure 5D&G&I*). These results indicated that the administration of FC-A may have the potential to enhance spastin's protein stability after spinal cord injury.

Enhancing the FC-A application could promote neurite outgrowth and regeneration suggesting that it can be used to promote neurite regrowth after injury in vivo. We, therefore, attempted to investigate whether FC-A could improve nerve regrowth after spinal cord injury in a contusion injury model. FC-A was administrated via intraperitoneal injection following SCI in mice. As shown in *Figure 5J*, *Figure 5—figure supplement 1*, H&E and LFB (Luxol fast blue stain) staining showed that the lesion site in the injury group presented the destroyed tissue structure with loss of myelination. Intriguingly, we found the tissue structure appeared to be normal and the area of demyelination in the lesion site strongly decreased in the FC-A group. In addition, in the spastazoline group, the lesion site appeared to be discontiguous, and the application of spastazoline reversed the protective effect of FC-A on spinal

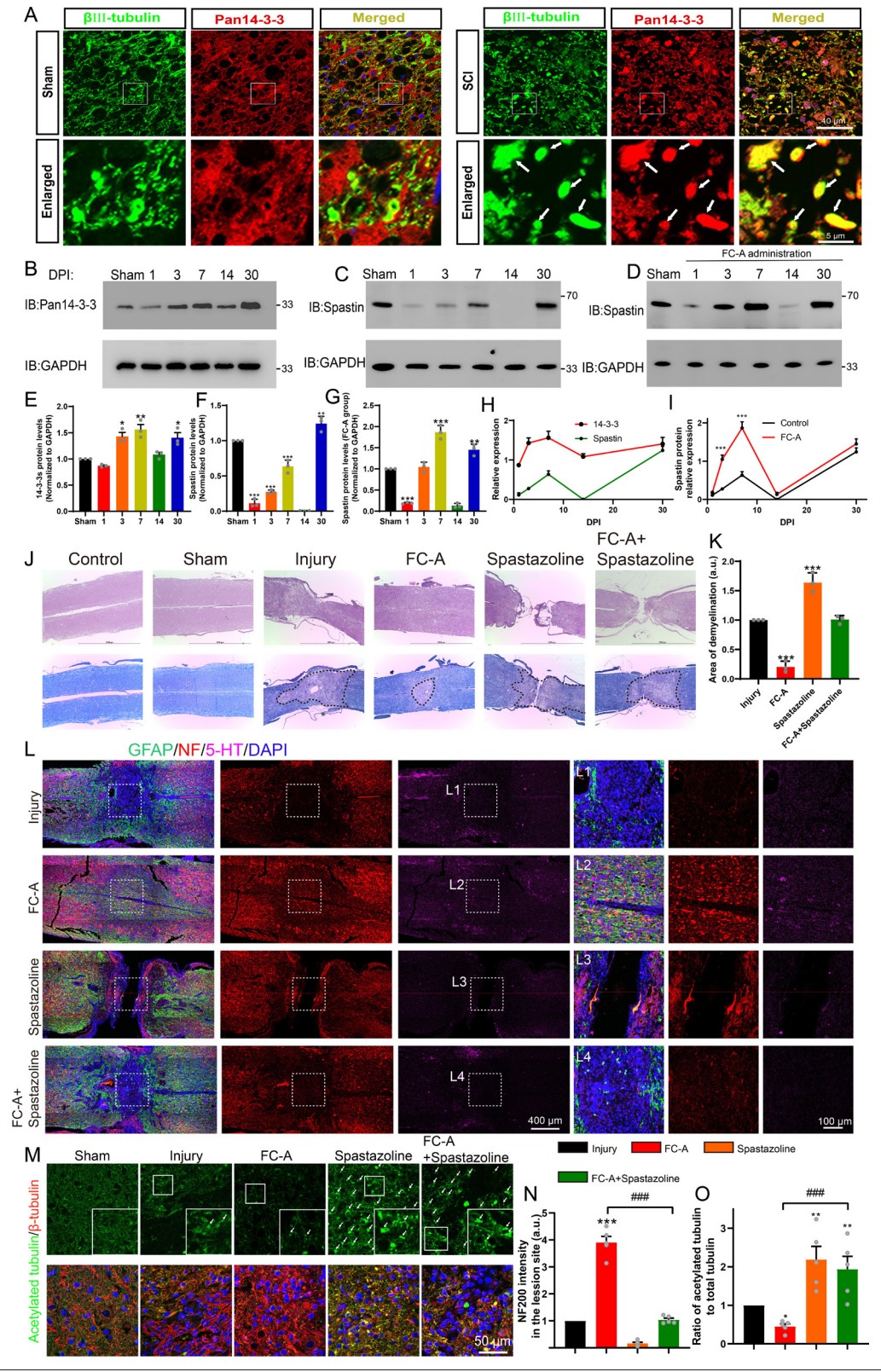

**Figure 5.** 14-3-3/spastin pathway is involved in nerve regeneration after spinal cord injury by targeting microtubules (MTs). (**A**) After spinal cord contusion, the lesion site of spinal tissue in the sham group and the SCI group were stained with Pan 14-3-3 (red) and βIII tubulin (green). The arrows mean the elevated 14-3-3 protein in the neuronal compartment. Scale bar: 40 μm, 5 μm. (**B**, **C**) After spinal cord contusion, the spinal cord tissues of the

*Figure 5 continued on next page*

*Figure 5 continued*

lesion site (near 1 cm) at indicated times (1, 3, 7 14, and 30 DPI) were ground and lysed, then subjected to western blot with 14-3-3 and spastin antibody. (**D**) After spinal cord injury, intraperitoneal injections of Fusicoccin-A (FC-A) were administered. Tissue from the lesion site was collected at indicated time points and subjected to western blot to observe the spastin protein levels. (**E, F**) Quantitative analysis of the 14-3-3 protein expression and spastin protein expression (n=3 per group). (**G**) Quantitative analysis of the spastin protein expression administrated with FC-A after spinal cord injury (n=3 per group). (**H**) Analysis of the expression trends of 14-3-3 and spastin proteins following spinal cord injury. (**I**) Trend analysis of spastin protein expression levels in lesion site of mice with spinal cord injury, comparing the injury control group with the FC-A treatment group. (**J**) 14-3-3 agonist FC-A and spastin inhibitor spastazoline were admisnistrated after spinal cord contusion. At 45 DPI (Days post-injury), the tissues were fixed and embedded, then cut horizontally and longitudinally. Hematoxylin and eosin (H&E) and LFB (Luxol fast blue stain) were stained. The spinal cord demyelination was shown inside the area of the dotted line. Scale bar: 2000 µm. (**K**) Quantitative analysis of demyelinated area in the lesion site among different groups (n=3, *p<0.05). (**L**) The slices were then subjected to immunofluorescence staining, and GFAP (green) was stained to label astrocytes, neurofilament (NF) (red) was stained to label neurons and 5-HT (magenta) was stained to label monoaminergic axons. The lesion site in different groups was boxed and enlarged in (L1, L2, L3, L4). Scale bar: 400 µm, 100 µm. (**M**) The spinal cord slices were stained with acetylated tubulin (Stable MTs which lack dynamics) and β-tubulin (total tubulin). Scale bar: 50 µm. (**N**) The normalized intensity of NF was quantified (n=5 animals per group). Mean ± SEM. *p<0.05, **p<0.01, ***p<0.001. (**O**) The normalized ratio of acetylation tubulin to total tubulin was calculated by the intensity value of acetylated tubulin divided by total tubulin (n=5 per group). Mean ± SEM. *p<0.05, **p<0.01, ***p<0.001.

The online version of this article includes the following source data and figure supplement(s) for figure 5:

**Source data 1.** Raw images for *Figure 5* (Part 1).

**Source data 2.** Raw images for *Figure 5* (Part 2).

**Source data 3.** Raw and annotated blots for *Figure 5*.

**Source data 4.** Numerical data for *Figure 5E–I*.

**Source data 5.** Numerical data for *Figure 5K*.

**Source data 6.** Numerical data for *Figure 5N–O*.

**Figure supplement 1.** After spinal cord contusion, the 14-3-3 agonist Fusicoccin-A (FC-A) and spastin inhibitor spastazoline were administered.

**Figure supplement 2.** Following spinal cord contusion, 14-3-3 agonist Fusicoccin-A (FC-A) and spastin inhibitor spastazoline were administered.

**Figure supplement 3.** Microtubule stability in the lesion site of mice after spinal cord contusion and administrated with Fusicoccin-A (FC-A) and spastazoline.

**Figure supplement 4.** Higher resolution image in *Figure 5—figure supplement 3*.

cord demyelination (*Figure 5K*). Since spastin plays a crucial role not only in neuronal development but also in mitosis, spastazoline may inhibit the proliferation of stromal cells in the lesion site, thereby impeding the wound-healing process following spinal cord injury. Next, we further investigated nerve regeneration by immunostaining with NF (Neurofilament) and 5-HT, which were co-stained with GFAP (astrocyte marker). As shown in *Figure 5E*, the border of the spinal cord lesion site was marked with astrogliosis (GFAP⁺), and the neurofilament and 5-HT immunogenicity in the lesion site were almost abolished compared to the uninjured area, indicating that the spinal cord contusion injury model was successfully established. Interestingly, in the FC-A group, we observed that the astrogliosis in the lesion site was attenuated and we found a great deal of neurofilaments and 5-HT signals in the lesion site (*Figure 5L*, boxed in L2), indicating that the application of FC-A enhanced the nerve regeneration after injury (Enlarged in *Figure 5—figure supplement 2*). In the spastazoline group, we also observed tissue discontinuity, which is consistent with *Figure 5J*, and almost no NF and 5-HT signals were found in the lesion site. Moreover, the application of spastazoline abolished the FC-A mediated-nerve regeneration after SCI. These results suggest that FC-A-mediated axonal regeneration after SCI requires the activation of spastin function.

In order to further confirm whether the intervention of 14-3-3 and spastin mediates nerve regeneration after spinal cord injury by targeting MTs at the site of spinal cord injury, acetylated tubulin, and β-tubulin were stained to label stale MTs (Lacking of dynamics) and total MTs. As shown in *Figure 5— figure supplements 3 and 4*, the intensity of acetylated microtubules was remarkably decreased in

the lesion site. Moreover, there was a clear boundary of acetylated MTs between the lesion site and the uninjured area. This may be ascribed to the activation of cell proliferation in the injured area after spinal cord injury which needs dynamic microtubules. Interestingly, we found the boundary of acetylated microtubules near the damaged area was unclear after the application of FC-A, and the intensity of acetylated microtubules significantly decreased compared with the injury group, indicating that microtubules tended to be in a dynamic state. However, after the application of spastazoline, the proportion of acetylated microtubules in the injury site increased significantly, and the boundary of acetylated microtubules in the injured area was unclear, indicating that the application of spastazoline led to a significant increase in the stability of the microtubules. In addition, the application of spastazoline can significantly reverse the enhancement of microtubule dynamics mediated by FC-A (*Figure 5M&O*). These experiments indicated that the administration of FC-A mediates nerve regeneration after spinal cord injury by affecting the MTs dynamics. In addition, these evidences also indicated that the FC-A and spastazoline can pass through the blood-brain barrier to reach the site of spinal cord injury.

## FC-A-mediated locomotor function improvement after SCI requires spastin participation

Since FC-A promotes nerve regeneration after spinal cord injury, we then investigated whether it could mediate the recovery of locomotor function after SCI. Therefore, we performed several experiments to verify whether the application of FC-A could affect the locomotor function recovery after SCI (*Figure 6A*). Catwalk analysis showed that the maximum contact area of the mice in the FC-A group was significantly larger than that in the injury group, and this phenomenon was reversed after the application of spastazoline (*Figure 6B&E*). The regularity index (Calculated as the number of normal strides multiplied by four and then divided by the total number of strides; this value is ~100% in normal animals) also had a similar phenomenon. The gait of the hindlimbs of mice in the FC-A group was more coordinated than that of the mice in the injury group (*Figure 6C*), and the regularity index was statistically enhanced compared with the injury group (*Figure 6G*). This phenomenon was also significantly suppressed after the application of spastazoline. We further evaluated the motor function after spinal cord injury by BMS (Basso mouse scale) score (*Figure 6D*). BMS is a 9-point scale that provides a gross indication of locomotor ability and determines the phases of locomotor recovery. We found that the BMS scores of the mice in the FC-A group was significantly improved compared with the mice in the injury group, and peaked after 35 days, while spastazoline reversed the improvement of locomotor function. At the same time, we used the footprint test to analyze the motor function of the mice (*Figure 6F*). The experimental data showed that the paw trailing phenomenon of the hindlimbs in the FC-A group was significantly improved compared with the injury group, and the stride length was significantly increased. This improvement was also significantly suppressed by the application of spastazoline (*Figure 6H*). Finally, we attempted to verify whether the spinal neural circuits in mice were remodeled by examining the motor-evoked potentials (Stimulation at the sensorimotor cortex in the M1 region and receive signals at the site of gastrocnemius). Our experimental results showed that the MEP current amplitude decreased significantly after spinal cord injury, while the current amplitude in the FC-A group significantly increased compared with the injury group. And the improvement effect was significantly weakened after the application of spastazoline, indicating that FC-A mediates the remodeling of neural circuits and normal spastin function is the prerequisite for the repair after spinal cord injury.

## FC-A mediates the repair of SCI in a T10 lateral hemisection model requires spastin participation

We have clarified the important role of 14-3-3 and spastin in the repair of spinal cord injury in the contusion model. In order to further observe the axon regeneration after spinal cord injury, we performed a spinal right lateral hemisection at the T10 level to eliminate the right side of axonal projections. Neurotransmission of serotonin (5-HT) in the spinal cord is required for the modulation of sensory, motor, and autonomic functions and can influence the response to spinal cord injury (*Perrin and Noristani, 2019*; *Yuan et al., 2017*). In addition, studies have reported that the 5-HT fibers have strong regenerative capacity and are involved in the regulation of neuronal activity (*Yuan et al., 2017*). Thus, we explored the 5-HT immunoactivity in spinal cord tissues isolated on 17 DPI. In the control

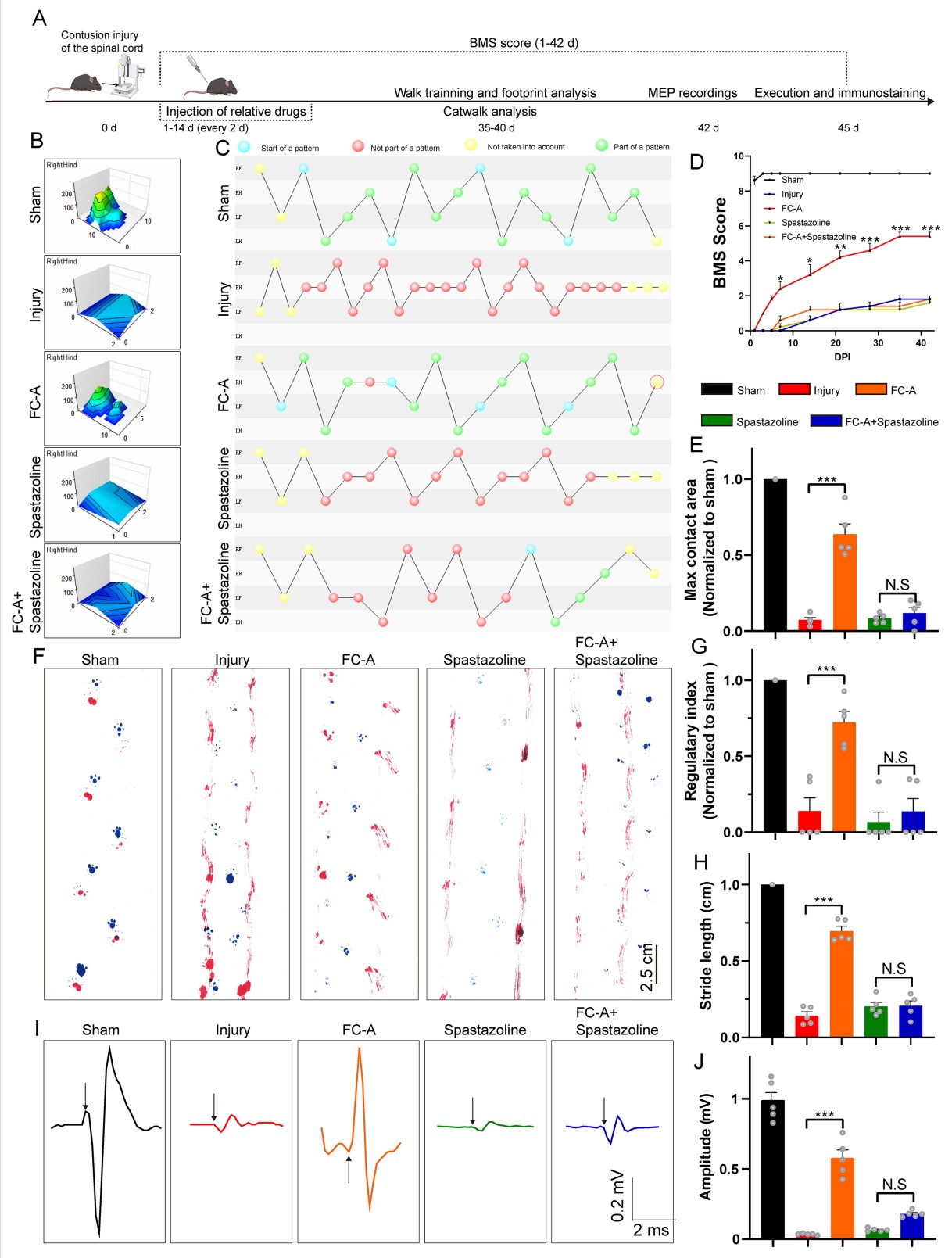

**Figure 6.** 14-3-3/spastin pathway coordinates locomotor recovery after spinal cord contusion. (**A**) Schematic illustration of relative treatments and examinations at different indicated times. (**B**) The max contact areas of the right hindlimb were recorded and analyzed by the Catwalk software. (**C**) The footfall patterns were visualized by the Catwalk software. The blue dot represents the start of a pattern. The red dot represents the gait was not part of a pattern. The yellow dot represents that the gait was not taken into account. The green one means the gait belongs to a normal part of a pattern.

*Figure 6 continued on next page*

*Figure 6 continued*

(**D**) Basso mouse scale (BMS) scores at different indicated days after injury and data were represented as mean ± SEM. (n=5 group, data were analyzed by one-way ANOVA, and Bonferroni's multiple-comparison test were used for post hoc comparisons). (**E**) The quantitative analysis of the hindlimb (right) max contact area. (n=5 animals per group). (**F**) Footprint analysis of the mice at 6 weeks after spinal cord contusion. The forelimbs were marked with blue and the hindlimbs were marked with red. Scale bar: 2.5 cm. (**G**) The normalized regularity index of the footfall pattern calculated by Catwalk software (~100% in the sham group, n=5 animals per group). (**H**) Quantitative analysis of the stride length in the footprint assay. The mean stride length of each walk was counted, n=5 animals per group. (**I**) Motor evoked potentials (MEPs) were recorded at 7 weeks after spinal cord contusion. The craniotomy was performed and the M1 region of the sensorimotor cortex were stimulated by a single square-wave stimulus of 0.5 mA, 0.5 ms duration, 2 ms time delay, and 1 Hz. The MEP were recorded with the signals detected by the electrode on the gastrocnemius muscle. The representative histograms of the amplitude-times are shown. The arrows indicated the stimulus. (**J**) The current amplitudes of the MEP were quantified, n=5 animals per group. Mean ± SEM. *p<0.05, **p<0.01, ***p<0.001.

The online version of this article includes the following source data for figure 6:

**Source data 1.** Numerical data for *Figure 6*.

SCI group, 5-HT-positive axons were significantly upregulated in the injury site, and to some extent in the rostral site. Interestingly, we found much stronger 5-HT staining in the lesion site of FC-A group and rostral site of injury compared with the SCI control group, indicating that the FC-A promoted the regeneration of 5-HT positive axons in vivo. However, few 5-HT signals were found in the lesion site, and also in the rostral site in the spastazoline group. In addition, the FC-A promoting effects of 5-HT axonal regeneration were significantly weakened by the combined application of spastazoline (*Figure 7A&C*).

We further investigated the immunoactivity of neurofilament and myelin basic protein (MBP) in the white matter of the lesion site to evaluate axonal regeneration and the extend of remyelination. Neurofilaments are cytoskeletal proteins that are expressed abundantly in the cytoplasm of axonal fibers in the CNS (*Liu et al., 2019*). The MBP forms and maintains the structure of the compact myelin sheath which then regulates axonal function (*Ceto et al., 2020*). Our results revealed that FC-A could enhance the regeneration of neurofilaments which were largely colocalized with MBP, whereas spastazoline alleviated this effect (*Figure 7B&D*). Collectively, these results suggest that FC-A promotes axonal regeneration which is dependent on activation of spastin microtubule severing.

Previous studies have reported that the proliferation of neural stem/progenitor cells is also reactivated and contributes to synapse formation following spinal cord injury (*Venkatesh et al., 2019*; *Ko et al., 2020*). Since spastin is a microtubule severing enzyme involved in cell division. Thus, we explored the possibility of whether 14-3-3/spastin complex participates in the proliferation process of neural stem/progenitor cells. Therefore, spinal cord tissues from the lesion site were stained with Nestin, BrdU, and NeuN. Intriguingly, we found that FC-A treatment after SCI significantly increased the expression of Nestin (*Figure 7—figure supplement 1*) and BrdU (*Figure 7—figure supplement 2*), and these effects were abolished by spastazoline treatment. These results demonstrated that the FC-A administration contributed to the proliferation of neural stem/progenitor cells thereby improving recovery after spinal cord injury.

We also further examined whether FC-A administration altered the motor function after SCI in this lateral hemisection injury model. Mice with SCI in all groups exhibited paralysis in the right hindlimb after hemisection which confirmed the successful establishment of the lateral hemisection SCI model. For mice in the SCI control group, the BMS scores gradually improved to ~4 by 16 DPI. In contrast, the scores of mice in the FC-A group indicated an improvement in locomotor function which was impaired in the spastazoline group. Moreover, the improvement in locomotor function following the administration of FC-A diminished when spastazoline was administered (*Figure 7E*). We further analyzed the footprint of mice with SCI and calculated the length or width of strides (*Figure 7—figure supplement 3*). We found that stride length improved in mice of the FC-A group but it was reduced in the spastazoline group although not significantly. The improvement induced by FC-A diminished when co-treated with spastazoline (*Figure 7G*). Notably, there was no significant difference in stride width between the groups (*Figure 7H*). Similar effects were observed in the foot fault scan experiment. The faults of the right hindlimb significantly decreased in the FC-A group and increased in the spastazoline group (*Figure 7F*). The improvement of hindlimb fault in mice administered with FC-A diminished following the application of spastazoline. We also explored whether the stability of MTs was altered by the administration of FC-A and spastazoline in the spinal cord injury model. We found that the ratio of

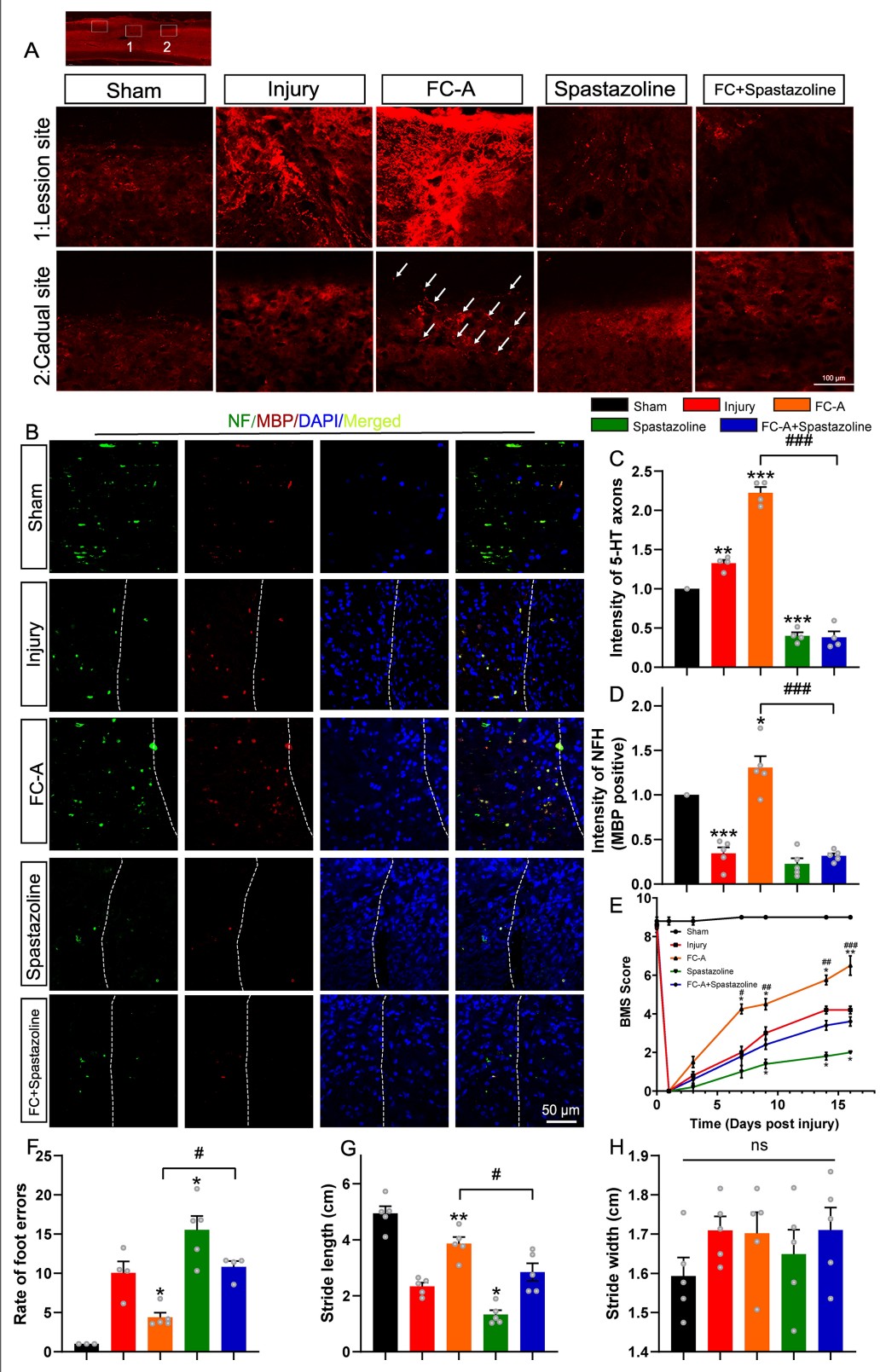

**Figure 7.** Fusicoccin-A (FC-A) promotes axon regeneration in a T10 lateral hemisection spinal cord injury model. (**A**) The spinal cords of adult mice were subjected to lateral hemisection and administrated by intraperitoneal injection with FC-A or spastazoline. Sagittal view of 5-HT-immunoreactive raphespinal fibers in the spinal cord on 17 DPI. Scale bar: 100 μm. (**B**) Sagittal view of neurofilaments which was myelin basic protein (MBP) positive

*Figure 7 continued on next page*

*Figure 7 continued*

in the white manner of the lesion site of the spinal cord. Scale bar: 50 μm. (**C**) The normalized quantification of 5-HT immunoreactive fluorescence intensity (0.5 mm caudal to the lesion site, n=5 animals per group). (**D**) The normalized quantification of NF (Neurofilament) immunoreactive fluorescence intensity in the lesion site (n=5 animals per group). (**E**) The locomotor function of mice after SCI were scored at the indicated time points according to the standard of the Basso Mouse Scale (BMS). (**F**) Foot fault test was performed and the total steps and steps dropped out of the right hindlimb were recorded. The rate of the foot errors was analyzed, n=5 animals per group. *p<0.05, **p<0.01, ***p<0.001. (**G, H**) The footprint assay was performed and stride length (**G**) and stride width (**H**) quantitatively analyzed. Data were presented as mean ± SEM, n=5 animals per group, the mean value of each animal's walk were calculated.

The online version of this article includes the following source data and figure supplement(s) for figure 7:

**Source data 1.** Raw Images for *Figure 7*.

**Source data 2.** Numerical data for *Figure 7C*.

**Source data 3.** Numerical data for *Figure 7D*.

**Source data 4.** Numerical data for *Figure 7E*.

**Source data 5.** Numerical data for *Figure 7F*.

**Source data 6.** Numerical data for *Figure 7G*.

**Source data 7.** Numerical data for *Figure 7H*.

**Figure supplement 1.** 14-3-3 agonist Fusicoccin-A (FC-A) promotes the Nestin expression which was NeuN-positive.

**Figure supplement 2.** 14-3-3 agonist Fusicoccin-A (FC-A) promotes the Nestin expression which is mainly co-localized with BrdU.

**Figure supplement 3.** Footprint assay of mice after the intervention by Fusicoccin-A (FC-A) and spastazoline following spinal cord injury.

**Figure supplement 4.** Microtubule stability of in the lesion site of mice in a T10 lateral hemisection spinal cord injury model after relative drug administration.

---

acetylated tubulin to total tubulin was significantly increased in the SCI group, decreased in the FC-A group, and upregulated in the spastazoline group, which further confirmed the successful delivery of drugs to the lesion site (*Figure 7—figure supplement 4*). These findings suggested that FC-A was effectively delivered to the spinal cord tissue and improved the locomotor function of mice with SCI, and spastin activation is the prerequisite the repair after spinal cord injury.

## Discussion

The limited cell intrinsic regrowth capability at the injury site of CNS damage or disease results in failure of axonal regeneration. Recent studies have demonstrated that the remodeling of microtubule dynamics was necessary for promoting axon regeneration (*Griffin and Bradke, 2020*; *Sarbanes et al., 2022*). Spastin is a MT severing enzyme which cuts long MTs into short fragments for local MTs remodeling and thus could be administrated to promote axon regeneration (*Lin et al., 2019*). In this study, we aimed to reveal the regulatory molecular mechanisms of spastin and its role in the recovery of spinal cord injury. We found that 14-3-3 interacted with spastin by targeting the phosphorylation of Ser233, protecting spastin against ubiquitin-mediated degradation, thus up-regulated the spastin-dependent microtubule severing activity. Moreover, the 14-3-3 agonist FC-A promoted the neurite outgrowth and regeneration, and normal spastin function activation is the prerequisite for the repair effects. In adult mice with of SCI (both in contusion and lateral hemisection spinal cord injury model), we found that FC-A promoted the recovery of locomotor function and axonal regeneration. Therefore, we postulated that the 14-3-3 and spastin is an attractive target for improving axonal regeneration and locomotor recovery after SCI.

To date, the severing activity induced by spastin has been well characterized (*Vemu et al., 2018*; *Costa and Sousa, 2022*) and reported to be essential in the formation of axon branches and axonal extension (*Jiang et al., 2020*; *Zhang et al., 2007*). Studies have also shown that spastin regulates axon regeneration in a dose-dependent manner. Of note, during regeneration, the cytoskeleton undergoes remodeling, suggesting that manipulation of MTs dynamics by spastin

may be a potential approach for controlling neural regeneration. However, efficient regeneration is dependent on the precise spastin dosage. Therefore, understanding the underlying regulatory mechanism of spastin is pivotal for identifying therapeutic targets in CNS. In this study, LC-MS-based proteomics analysis was performed and proteins that bind spastin were screened to investigate the regulatory mechanisms of spastin during CNS damage (*Figure 1A*, *Figure 1—figure supplement 1*). Among the identified binding proteins, 14-3-3 proteins had high scores and were found to interact with spastin endogenously in the spinal cord (*Figure 1B and C*). Of note, MS data from a previous study also showed that the spastin peptides were present in the binding peptides with 14-3-3 agonist Fusicoccin (*Lin et al., 2019*), which confirmed the important relevance between these two proteins. Considering that both 14-3-3 and spastin were found to regulate axon outgrowth and regeneration, the localizations of both proteins were characterized within hippocampal neurons. In addition, GST pulldown and co-immunoprecipitation assays further verified the interaction between all 14-3-3 isoforms and spastin (*Figure 1F–H*). Also, the binding between 14-3-3 and spastin was further characterized which showed that Ser233 in spastin was responsible for the binding, and could directly influence the fate of neurite outgrowth (*Figure 2*). Previously, it was found that spastin was phosphorylated at Ser233 (human Ser268) by HIPK2 and this process was vital for successful abscission and the loss of HIPK2 exhibited a protective effect on neurons (*Cornell et al., 2016*). Our western blotting results showed that HIPK2 was expressed at 9[th] DIV in the brain tissue (not presented), suggesting that there may be other kinases that regulate Ser233 phosphorylation during neural development. Therefore, we have well-established the model of the 14-3-3/spastin binding which could be the intervention target for regulating the MT severing mediated by spastin.

The regulatory mechanisms that control the MT severing activity mediated by spastin are largely unknown. In our previous study, spastin protein expression levels and its severing effect were regulated at post-transcriptional levels, such as micoRNA-30 (*Jiang et al., 2020*) and SUMOylation (*Ji et al., 2020*). In this study, we found that 14-3-3 interacted with spastin from the perspective of protein-protein interactions, and we showed that 14-3-3 prevented the ubiquitin-mediated degradation of spastin and upregulated the spastin protein levels by inhibiting the ubiquitin pathway and phosphorylation of Ser233 was crucial in this process. Our results showed that Ser233 de-phosphorylation lowered the protein stability, consistent with findings from a recent study (*Sardina et al., 2020*), and our study further illustrated the molecular mechanisms of this phenomenon. In addition, we observed that 14-3-3 up-regulated the microtubule severing activity of spastin by interacting with its phosphorylation at Ser233. Based on this mechanism, we proposed a model in which Ser233 is phosphorylated during the binding of 14-3-3 to spastin, thereby disrupting the spastin/ubiquitin binding, ultimately affecting the stability of spastin and subsequent neurite outgrowth and regeneration. Similar mechanisms were reported for doublecortin (*Dar et al., 2014*), cdt2 (*Lee and Lu, 2011*), p21 (*Chan et al., 2021*) and other proteins (*Kast and Dominguez, 2019*). For example, 14-3-3ε was found to interact with threonine 42 in doublecortin which protected it from degradation by inhibiting its ubiquitination. In this way, it regulated the doublecortin-regulated microtubule stability and neurite initiation process (*Lee and Lu, 2011*). In this part, we validated that 14-3-3 stabilized and upregulated the protein levels of spastin and enhanced the MT severing activity mediated by spastin which was similar to the mechanism of 14-3-3/cdt2 binding.

Since 14-3-3 proteins are adaptor proteins that are highly expressed in neural tissues, they are important regulators in neural development (*Cornell and Toyo-Oka, 2017*; *Marra et al., 2021*). Using a well-established 14-3-3/spastin interaction model, we found that promoting the MT dynamics by enhancing the interaction of 14-3-3/spastin could enhance neurite outgrowth and regeneration. This accelerated neurite outgrowth and repair following FC-A application by facilitating the binding of 14-3-3/spastin (*Figure 4*). In accordance with previous findings, 14-3-3 proteins can promote neurite outgrowth and axonal regeneration (*Kaplan et al., 2017*). In our study, we further demonstrated that enhanced 14-3-3/spastin interaction by FC-A favored neurite outgrowth and regeneration, but this effect was blunted by spastazoline treatment. Spastazoline was designed to compete with the AAA domain of spastin and this effectively abolished the severing activity mediated by spastin (*Pisa et al., 2019b*). The results also demonstrated that the normal severing activity mediated by spastin was required for neurite outgrowth and regeneration (*Figure 4*). These findings open a new window for developing spastin-based agents for accelerating axon regeneration after injury.

So far, few studies have investigated the relationship between spastin and neural regeneration in CNS diseases, such as spinal cord injury or brain injury. However, many studies have elucidated its indispensable role in the process of axon regeneration (*Lin et al., 2019*). In this study, we first observed a downregulation of spastin expression on 1st day post-spinal cord injury, which gradually increased in the recovery process. We also indicate that functional spastin is required for axon regeneration and locomotor recovery after spinal cord injury (*Figures 5 and 6*). Moreover, 14-3-3 was upregulated at 3 and 7 DPI, and was even higher at 30 DPI in the neuronal compartment, indicating that 14-3-3 may play an important role in the process of recovery after spinal cord injury (*Figure 5*). Using the model of 14-3-3/spastin interaction, we further elucidated the mechanism by which FC-A administration regulates the recovery of spinal cord injury by targeting MTs. (FC-A) is a fusicoccane, a small molecule produced by the *Phomopsis amygdali fungus* that stabilizes the binding of 14-3-3 to its substrates (*Liu et al., 1995*). The crystal structure of FC-A bound to 14-3-3 has already revealed the efficiency of the enhancement of FC-A in the interaction between 14-3-3 and their substrate (*Ji et al., 2018*). In addition, FC-A has been reported to promote axonal regeneration in mammals (*Kaplan et al., 2017*), although the mechanisms are rarely elusive. In this study, We found that FC-A can enhance the expression of spastin after spinal cord injury, and FC-A administration restored nerve regeneration and locomotor recovery after SCI (in both contusion and lateral hemisection spinal cord injury model), but these improvements were abolished following spastazoline administration (inhibiting the MT severing by direct competing with AAA domain of spastin *Pisa et al., 2019b*), suggesting the important role of 14-3-3 and spastin in the neural circuits remolding after SCI. This phenomenon was further confirmed by examining the ratio of acetylated tubulin in the lesion site of SCI, and our results showed the intervention of 14-3-3 and spastin by FC-A and spastazoline could significantly affect the MTs dynamics in the lesion site which indirectly demonstrated the efficient delivery of these drugs to the spinal cord across the blood-brain barrier. Since spastin plays an essential role in cell abscission, and the FC-A promoted the proliferation of neural stem/progenitor cells (*Figure 7—figure supplement 2*) thereby supporting axonal regeneration. These results also support that FC-A enhanced axonal regeneration after spinal cord injury, and the microtubule severing independent of spastin was important in this process. Therefore, our findings present 14-3-3 and spastin as a novel intervention for accelerating nerve regeneration after spinal cord injury.

In summary, we proposed the 14-3-3 and spastin binding model for manipulating microtubule dynamics and can be an effective intervention for promoting recovery of spinal cord injury. In addition, our results indicate that 14-3-3 and spastin are important regulators for nerve regeneration and 14-3-3 and spastin may be important targets for the treatment of spinal cord injury.

## Materials and methods

### Animals

Sprague Dawley rats (P0, male or female) were used for primary hippocampal and cortical neuron culture, whereas adult female mice (C57BL/6 J, 6–8 weeks) were used to establish the lateral hemitransection model and the contusion model of spinal cord injury. Animals were housed in a room, maintained at 22 °C under a 12:12 hr light/dark photoperiod, and given food and water ad libitum. All animal treatments were carried out in strict adherence to the guidelines by the National Institutes of Health's Guide for the Care and Use of Laboratory Animals. The study protocol was approved by the Jinan University Institutional Animal Care and Use Committee (Permit Number:20220705–03). The number of animals utilized and suffering were both reduced to the absolute minimum.

### Cell culture and transfection

Dissociated hippocampi were digested with 2 mg/mL papain and plated at a density of $1 \times 10^4$ cells/$cm^2$ onto poly-D-lysine-coated coverslips for primary hippocampal neuron culture. Cultures were kept alive in an incubator humidified with 5% $CO_2$ at 37 °C using Neurobasal-A media supplemented with 0.5 mM glutamine and 2% B27. Every three days, half of the cultural media was replaced. Various plasmids were transfected with calcium phosphate at 2nd day in vitro (DIV). COS7 cells (NCACC, Cat#GNO28) and human embryonic kidney (HEK) 293 cells (NCACC, Cat#SCSP-5209) were obtained from the National Collection of Authenticated Cell Cultures (NCACC) in China. The identity has been authenticated (STR profiling) and the tests for mycoplasma contamination were negative provided

by NCACC. Cell lines were grown in a 5% $CO_2$ incubator at 37 °C using Dulbecco's modified Eagle media with 10% fetal bovine serum and 1% penicillin/streptomycin as supplemental ingredients. The constructs were transfected into the COS7 cells and HEK293 cells using Lipofectamine 2000. Cells were cultured for 24–36 hr after transfection before being harvested.

## Plasmids and constructs

Several 14-3-3 isoforms were amplified in cDNA from Sprague Dawley rats via Polymerase Chain Reaction (PCR) as previously described *Ji et al., 2020*. The amplicons were inserted into the pCMV-Tag-2b or pGEX-5X-3 vector (Addgene). Spastin mutants were generated using the Quickchange Kit (Agilent, Santa Clara, CA). R18 was synthetized by IGE Biotechnology LTD (Guangzhou, China) and inserted into pCDNA 3.1 vector. All constructs were verified by sequencing by IGE Biotechnology LTD (Guangzhou, China). The following PCR primers were used:

| Vector | Primer | Sequence |
|---|---|---|
| pGEX-5X-3 | *Ywhab*-F | CCCAGGAATTCCCGGGTCGACATGACCATGGACAAAAGTGA |
| | *Ywhab*-R | TCAGTCAGTCACGATGCGGCCGCTTAGTTCTCTCCCTCTCCAG |
| pGEX-5X-3 | *Ywhae*-F | CCCAGGAATTCCCGGGTCGACATGGATGATCGGGAGGATCT |
| | *Ywhae*-R | TCAGTCAGTCACGATGCGGCCGCTCACTGATTCTCATCTTCCA |
| pGEX-5X-3 | *Ywhah*-F | CCCAGGAATTCCCGGGTCGACATGGGGGACCGAGAGCAGCT |
| | *Ywhah*-R | TCAGTCAGTCACGATGCGGCCGCTCAGTTGCCTTCTCCGGCTT |
| pGEX-5X-3 | *Ywhag*-F | CCCAGGAATTCCCGGGTCGACATGGTGGACCGCGAGCAACT |
| | *Ywhag*-R | TCAGTCAGTCACGATGCGGCCGCTTAGTTGTTGCCTTCGCCGC |
| pGEX-5X-3 | *Ywhaz*-F | CCCAGGAATTCCCGGGTCGACATGGATAAAAATGAGCTGGT |
| | *Ywhaz*-R | TCAGTCAGTCACGATGCGGCCGCTTAATTTTCCCCTCCTTCTC |
| pGEX-5X-3 | *Ywhaq*-F | GGAATTCCCGGGTCGACATGGAGAAGACCGAGCTGATCCAGA |
| | *Ywhaq*-R | TCAGTCACGATGCGGCCGCTTAGTTCTCAGCCCCCTCAGCTGC |
| pCMV-Tag-2b | *Ywhab*-F | CTTGGTACCGAGCTCGGATCCGCCACCATGACCATGGACAAAAGTGA |
| | *Ywhab*-R | GCCCTTGGACACCATGGATCCTTAGTTCTCTCCCTCTCCAG |
| pCMV-Tag-2b | *Ywhae*-F | CTTGGTACCGAGCTCGGATCCGCCACCATGGATGATCGGGAGGATCT |
| | *Ywhae*-R | GCCCTTGGACACCATGGATCCTCACTGATTCTCATCTTCCA |
| pCMV-Tag-2b | *Ywhah*-F | CTTGGTACCGAGCTCGGATCCGCCACCATGGGGGACCGAGAGCAGCT |
| | *Ywhah*-R | GCCCTTGGACACCATGGATCCTCAGTTGCCTTCTCCGGCTT |
| pCMV-Tag-2b | *Ywhag*-F | CTTGGTACCGAGCTCGGATCCGCCACCATGGTGGACCGCGAGCAACT |
| | *Ywhag*-R | GCCCTTGGACACCATGGATCCTTAGTTGTTGCCTTCGCCGC |
| pCMV-Tag-2b | *Ywhaz*-F | CTTGGTACCGAGCTCGGATCCGCCACCATGGATAAAAATGAGCTGGT |
| | *Ywhaz*-R | GCCCTTGGACACCATGGATCCTTAATTTTCCCCTCCTTCTC |
| pCMV-Tag-2b | *Ywhaq*-F | CTTGGTACCGAGCTCGGATCCGCCACCATGGAGAAGACCGAGCTGATCC |
| | *Ywhaq*-R | GCCCTTGGACACCATGGATCCTTAGTTCTCAGCCCCCTCAGCTGC |
| | *Spast* (S233A)-F | CACCACAGGGCGCCTGCCTGCAGTGGTTTATCC |
| | *Spast* (S233A)-R | GGATAAACCACTGCAGGCAGGCGCCCTGTGGTG |
| | *Spast* (S562A)-F | AAAAGATCAAACGCAGTGTGGCCCCTCAGACTTTAGAAGCATA |
| | *Spast* (S562A)-R | TATGCTTCTAAAGTCTGAGGGGCCACACTGCGTTTGATCTTTT |
| | *Spast* (S233D-F) | CACCACAGGGCGCCTGATTGCAGTGGTTTATCC |

*Continued on next page*

*Continued*

| Vector | Primer | Sequence |
| --- | --- | --- |
| | *Spast* (S233D-R) | GGATAAACCACTGCAATCAGGCGCCCTGTGGTG |
| pEGFP-C1 | *Spast*-F | TACAAGTCCGGACTCAGATCTGCCACCATGAGTTCTCCGGCCGGACG |
| | *Spast*-R | GTACCGTCGACTGCAGAATTCTCATTAAACAGTGGTGTCTCCAA |
| pCMV-Tag-2b | *Ubb*-F | CGATAAGGCCCGGGCGGATCCATGCAAATCTTCGTGAAGAC |
| | *Ubb*-R | GATAAGCTTGATATCGAATTCTTAATAGCCACCCCTCAG |

## Expression of recombinant proteins and GST-Pulldown assay

Purification of GST fusion of 14-3-3 proteins and pulldown experiments were performed as previously described *Ji et al., 2020*; *Patel et al., 2021*. Briefly, GST-14-3-3s constructs (Invitrogen) were transformed into *Escherichia coli* strain BL21 (DE3) strains. Isopropy-D-thiogalactoside was incubated for 6 hr at 30 °C to stimulate the synthesis of fusion proteins. The bacteria were first centrifuged, and then resuspended in a mixture of protease inhibitors (Merck, Whitehouse Station, NJ). The cell suspension was treated with 0.1% lysozyme, followed by 0.5% deoxycholic acid on ice for 20 min. The contents were, sonicated (10,000 rpm for 30 min), centrifuged and the supernatant collected, then Glutathione-Sepharose beads (supplemented with 1% Triton X-100) used to purify GST fusion proteins in the supernatant. Next, 5 µg of GST-fusion protein was incubated with 400 µg of protein from the spinal cord (1 cm around T10) of SD rats (P0) under gentle rotation, then centrifugated to collect the GST-binding proteins. Western blot assay was then performed to quantify GST-binding proteins.

## Immunoprecipitation and western blotting

Immunoprecipitation (IP) assays were carried out as previously described *Ji et al., 2021*. Briefly, total cell extracts or tissues (1 cm around T10) were prepared by soaking them in 400 µL of lysis buffer for 30 min at 4 °C. The lysates were centrifuged at 10,000 rpm for 10 min at 4 °C after a brief sonication. The extract was immunoprecipitated with 2 µg antibodies against spastin, GFP, or Flag, and then incubated for 4 hr at 4 °C with 60 µL of protein G plus protein A/G agarose using continuous gentle inversion. Three times, the immune complexes were pelleted and washed. Western blot assay was performed to analyze the precipitated complexes. Western blot analysis was carried out as previously described *Ji et al., 2020*. SDS-PAGE was used to separate the lysates, which were then electrophoretically transferred to a polyvinylidene difluoride membrane. Membranes were probed with primary antibodies at 4 °C overnight after being blocked in Tris-buffered saline with 5% milk and 0.05% Tween. The membranes were cleaned, incubated with goat anti-mouse or anti-rabbit secondary antibodies that were horseradish peroxidase-conjugated, and then visualized with ECL-prime. Anti-spastin (1:1000, ABN368 from Millipore), anti-Pan 14-3-3 (1:1000, #8312 from Cell Signaling Technology), anti-Pan phosphoserine/threonine (Cat#612548) form BD life science, anti-GFP (1:5000, Ab290 from Abcam), anti-GFP (1:10000, 50430–2-AP from Proteintech), anti-Flag (1:1000, F3165 from Sigma-Aldrich), anti-GAPDH (1:5000, AC002 from ABclonal), and HRP conjugated goat-derived anti-rabbit or anti-mouse were the antibodies used (1:5000, Abclonal).

## Scratch assay

Rat cortical neurons were seeded in 96-well plates (30,000 cells per well), and then a plastic P10 pipet tip was used to scratch across the center of the well. Half the media was aspirated out and replaced with fresh ones every three days until the 7th DIV. The wells were immediately filled with chemicals after scratching. The cultures were fixed and stained with anti-βIII tubulin (1:1000, ab18207) the following day. Images of axons stained with βIII tubulin were collected, imported in ImageJ, and the proportion of neurites stained by βIII tubulin in the central 70% of the scratch was calculated.

## Lateral hemi-sectioning and contusion model of SCI

The procedure for T10 lateral hemi-sectioning and contusion model was as previously described (*Liu et al., 2017*; *Brown and Martinez, 2018*). Briefly, mice were first anesthetized with 1.25% 2,2,2-tribromo ethanol (0.2 mL/100 g), via intraperitoneal injection for 3–4 min, then a laminectomy

performed near T10 after making a midline incision over the thoracic vertebrae. The T10 level of the spinal cord was recognized by the anatomical landmark which is parallel to the twelfth rib of the mice. For the hemi-sectioning injury model, the right unilateral hemi-section was carefully performed using a scalpel and microscissors, with care taken to avoid damaging the spinal cord dura. For the contusion model, the nitrogen tank controlling the impactor tip was set at 18 psi or 124 kPa. A U-shaped stabilizer with the rat was loaded onto the stage of the Louisville Injury System Apparatus (LISA) and the dura of the spinal cord was adjusted directly under the impactor while monitored by the laser beam. Crash depth was adjusted to 1.0 mm for injury, and with time set to 0.5 s. Finally, the muscles were stitched and the skin was stapled back together. In this study, exclusion criteria were applied to mice exhibiting inaccuracies in the hemisection model concerning the incision's accurate placement with respect to the posterior median vein.

## Pharmacological interference with FC-A or spastin

After the establishment of the spinal cord injury model, the mice within the injury group underwent random allocation into cages following a double-blind methodology. Another experimenter, blinded to the groupings, then administered the indicated drug treatment randomly to the mice within the injury group. An equal volume of normal saline was intraperitoneally injected into SCI control mice. Mice were intraperitoneally injected with FC-A (Enzo life sciences, ALX-350–115) at a dose of 15 mg/kg starting immediately after SCI and every other day until the end of the experiment. Mice were intraperitoneally injected with spastazoline (MedChemExpress, HY-111548) at a dose of 20 mg/kg, immediately after SCI then every other day until the end of experiments.

## Basso mouse scale

Two observers who were blind to the experimental groups conducted the Basso Mouse Scale (BMS) after the T10 SCI in accordance with a previously described procedure (*Sun et al., 2018*). Mice were briefly left in an open field for 4 min to observe all visually perceptible signs of locomotor recovery. The scores were based on hind limb movements performed in an open field and included hind limb joint movement, weight support, plantar stepping, coordination, paw position, and trunk and tail control (0, full hind limb paralysis; 9, normal locomotion).

## Catwalk analysis

According to previously described methods (*Zhou et al., 2020*), the CatWalk XT 9.1 automated quantitative gait analysis system (Noldus Company, Netherlands) was also used to evaluate the locomotor recovery of mice after injury. Briefly, one week before surgery, mice were trained to walk in a single direction for the entire length of a darkened Catwalk chamber in a quiet room five times daily. Seven weeks after surgery, the mice were assessed five times using the same system. Gait regularity index and hindlimb contact area were recorded. The regularity index was calculated as the number of normal strides multiplied by four and then divided by the total number of strides; this value is ~100% in normal animals.

## Footprint analysis

Footprint analysis was conducted before the mice were executed, using a previously described protocol (*Kamens and Crabbe, 2007*). Briefly, each animal's hindlimbs were first coated with nontoxic paint. The mice were then moved from a brightly illuminated starting box to a darkened one, by allowing them to walk through a narrow-custom-built plexiglass trough (5 cm wide by 40 cm long), during which they left a trace of their paw prints on the white sheet that overlaid the trough. The footprints were scanned, and digitized images were measured using Image J Pro Plus. Stride length was measured as the distance between the adjacent hindlimbs, whereas stride width was taken as the distance between the affected and the contralateral limbs. At least ten footprints per side, from three sessions per animal, were measured for both parameters and their average were calculated.

## Foot fault test

The Parallel Rod apparatus was used in this test (*Kamens and Crabbe, 2007*). Summarily, mice were positioned in the middle of the device, and allowed to walk for 3 min during which a video was captured. The test was performed by two observers who were blinded to the experimental groups.

One observer recorded the total number of steps taken, while the other counted instances where each right hind limb slipped through the rod. Three runs were recorded for every mouse.

## Motor evoked potential (MEP) recordings

To assess SCI recovery, MEPs were recorded 7 weeks after SCI according to previously described methods (*Zhou et al., 2020*). First, mice were anesthetized with 1.25% 2,2,2-tribromo ethanol (0.2 mL/100 g). A craniotomy was then performed to expose the M1 region of the motor cortex. Electrode penetration was guided via a stereotaxic instrument to a depth of 700–1000 mm from the brain surface to target corticospinal neurons in the sensorimotor cortex. The recording electrode was placed on the gastrocnemius muscle and the reference electrode was placed on the paraspinal muscle between the stimulation and recording points. A ground electrode was attached to the tail. A single square-wave stimulus of 0.5 mA, 0.5 ms duration, 2 ms time delay, and 1 Hz was used. Amplitude was measured from the beginning of the first response wave to its maximum point. All potentials were amplified and acquired using a digital oscilloscope (Chengdu Instrument Factory, Chengdu, China).

## Immunocytochemistry

Cells were grown on coverslips and processed according to immunocytochemistry protocols as previously described *Ji et al., 2020*. Cells were fixed using freshly prepared 4% paraformaldehyde, followed by permeabilization with 0.1% Triton X-100 in TBS and blocking in 3% normal donkey serum. The cells on the coverslips were incubated with anti-GFP (1:1000, Ab290 form Abcam) or anti-α-tubulin (1:1000, ab7291 from Abcam) and then incubated with secondary antibody (Donkey anti-Rabbit IgG (H+L) highly cross-adsorbed secondary antibody-Alexa Fluor 488, RRID:AB_2535792. Donkey anti-Mouse IgG (H+L) highly cross-adsorbed secondary antibody, Alexa Fluor 555, RRID:AB_2762848, Donkey anti-Mouse IgG (H+L) highly cross-adsorbed secondary antibody, Alexa Fluor 647, RRID:AB_2866490). The coverslips were then mounted with DAPI (diamidino-2-phenylindole)-free Fluoro-Gel on glass slides (Electron Microscopy Sciences, USA). Cells were scanned using a 63 x oil immersion objective mounted on a confocal microscope (LSM 710 Meta; Carl Zeiss). LSM 710 software was used to process the images after they were captured using sequential acquisition settings at a resolution of 1024×1024 pixels and a 12-bit depth.

Mice were administered consecutive transcardial infusions of 4% paraformaldehyde in PBS and 0.09% saline after being given a deep anesthetic. After tissue collection, post-fixation in 4% paraformaldehyde was applied overnight, and tissues were then immersed in 30% sucrose for two days to dehydrate them. Serial sagittal cryostat-sections were incubated at 4 °C for at least 18 hr before being stained with primary antibodies in blocking buffer (3% goat serum and 5% donkey serum in 0.3% PBST). Alexa Fluor-coupled secondary antibodies (Life Technologies) diluted in blocking buffer were incubated for 2 hr at room temperature to perform the detection. Anti-GFAP (1:1000, GB11096 from Servicebio), anti-serotonin (1:5000, AB125 from Sigma-Aldrich), anti-neurofililament (1:200, A19084 from ABclonal), anti-Myelin Basic Protein (1:200, A11162 from ABclonal), anti-BrdU (1:200, A20790 from ABclonal), anti-Nestin (1:200, A11861 from ABclonal), and anti-NeuN (1:200, A19086 from ABclonal) were used. Images were scanned using a confocal microscope.

## Statistical analysis

The analyses were carried out by GraphPad 8.4.0 software. Differences across multiple groups were determined using One-way analysis of variance (ANOVA), followed by Newman-Keuls post hoc tests for mean separations. $p \leq 0.05$ were considered statistically significant.

# Acknowledgements

This work was supported by the Natural Science Foundation of China (grant nos. 82102314, 82372504 and 32170977), the Guangdong Basic and Applied Basic Research Foundation (grant nos. 2022A1515010438 and 2022A1515012306), Science and Technology Projects in Guangzhou (grant nos. 2023A04J1284, 2023A03J1024 and 202201020018), the Clinical Frontier Technology Program of the First Affiliated Hospital of Jinan University, China (no. JNU1AF- CFTP- 2022- a01206), and project funded by China Postdoctoral Science Foundation (2023M731320).

# Additional information

## Funding

| Funder | Grant reference number | Author |
| --- | --- | --- |
| National Natural Science Foundation of China | 82102314 | Zhisheng Ji |
| National Natural Science Foundation of China | 82372504 | Hongsheng Lin |
| National Natural Science Foundation of China | 32170977 | Hongsheng Lin |
| Guangdong Basic and Applied Basic Research Foundation | 2022A1515010438 | Zhisheng Ji |
| Guangdong Basic and Applied Basic Research Foundation | 2022A1515012306 | Hongsheng Lin |
| Scientific and Technological Planning Project of Guangzhou City | 2023A04J1284 | Zhisheng Ji |
| Scientific and Technological Planning Project of Guangzhou City | 2023A03J1024 | Hongsheng Lin |
| Scientific and Technological Planning Project of Guangzhou City | 202201020018 | Hongsheng Lin |
| Jinan University | JNU1AF- CFTP- 2022-a01206 | Hongsheng Lin |
| China Postdoctoral Science Foundation | 2023M731320 | Qiuling Liu |

The funders had no role in study design, data collection and interpretation, or the decision to submit the work for publication.

## Author contributions

Qiuling Liu, Conceptualization, Resources, Data curation, Software, Formal analysis, Supervision, Funding acquisition, Validation, Investigation, Visualization, Methodology, Writing - original draft, Project administration, Writing – review and editing; Hua Yang, Data curation, Formal analysis, Validation, Investigation, Visualization, Methodology; Jianxian Luo, Conceptualization, Data curation, Formal analysis, Investigation, Methodology; Cheng Peng, Data curation, Formal analysis, Validation, Investigation, Methodology; Ke Wang, Data curation, Formal analysis, Investigation, Methodology; Guowei Zhang, Conceptualization, Data curation, Formal analysis, Investigation, Visualization, Methodology; Hongsheng Lin, Conceptualization, Formal analysis, Supervision, Funding acquisition, Validation, Visualization, Methodology, Project administration, Writing – review and editing; Zhisheng Ji, Conceptualization, Resources, Formal analysis, Supervision, Funding acquisition, Visualization, Methodology, Project administration, Writing – review and editing

## Author ORCIDs

Qiuling Liu http://orcid.org/0000-0002-8498-4646
Zhisheng Ji http://orcid.org/0000-0002-0404-855X

## Ethics

All animal treatments were carried out in strict adherence to the guidelines by the National Institutes of Health's Guide for the Care and Use of Laboratory Animals. The study protocol was approved by Jinan University Institutional Animal Care and Use Committee (Permit Number: 20220705-03). The number of animals utilized and suffering were both reduced to the absolute minimum.

Joint Public Review: https://doi.org/10.7554/eLife.90184.4.sa1
Author Response https://doi.org/10.7554/eLife.90184.4.sa2

## Additional files

### Supplementary files
• MDAR checklist

### Data availability
All data analysed during this study are included in the manuscript; source data files have been provided for all figures.

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
