## [Editor Report · eLife assessment]

The finding that Fusicoccin (FC-A) promotes locomotor recovery after spinal cord injury is supported by **solid** data, and the idea of harnessing small molecules that may affect protein-protein interactions to promote axon regeneration is **valuable**. The evidence showing that 14-3-3 and spastin interact and that 14-3-3 enhances spastin function and stability in cells is **solid**.

---

## [Referee Report · Joint Public Review]

The present work establishes 14-3-3 proteins as binding partners of spastin and suggests that this binding is positively regulated by phosphorylation of spastin. The authors show evidence that 14-3-3 - spastin binding prevents spastin ubiquitination and final proteasomal degradation. By using drugs and peptides that separately inhibit 14-3-3 binding or spastin activity, they show that both proteins are necessary for axon regeneration in cell culture and in vivo models in rats.

Major strengths

-The data establishing 14-3-3 and spastin as binding partners is convincing, as is its regulation by phosphorylation and its impact on protein levels related to the activity of the ubiquitin-proteasome system.

-The effects of FC-A on locomotor recovery after spinal cord contusion is very interesting.

Major weaknesses

-Given that spastazoline has a major impact on neurite outgrowth suggests that cells simply cannot grow in the presence of the inhibitor and raises serious questions about any selectivity for the concomitant effect FC-A - dependent growth.

-The histological data and analyses following spinal cord injury are not convincing. For example, the colabeling of NF and 5-HT is not convincingly labeling fibers. Also, the quality, resolution and size of the images is insufficient to support the quantitative data and it is hence difficult to interpret the data. Reviewers recognize that during the review process, efforts were made to improve the quality of the images.

-Reviewers also observed that the data to infer Spastin actions on Microtubules across different experimental models is weak and that claims about "MT severing" and "microtubules dynamics" were wrongly used given the provided evidence.

-The manuscript lacks direct evidence that a 14-3-3 and spastin function as a complex in the same pathway to promote regeneration. It is recognized, however, that the authors had made changes in the manuscript title and claims not to imply that the current evidence is sufficient in that matter.

---

## [Author Response]

The following is the authors’ response to the previous reviews.

**eLife assessment**
The finding that Fusicoccin (FC-A) promotes locomotor recovery after spinal cord injury is useful, and the idea of harnessing small molecules that may affect protein-protein interactions to promote axon regeneration is interesting and worthy of study. However, the main methods, data, and analyses are inadequate to support the primary claim of the manuscript that a 14-3-3-Spastin complex is necessary for the observed FC-A effects.

Response: We appreciate the eLife editorial and review team for consideration and evaluation of our manuscript. In light of the feedback from the editors and reviewers, we recognize that certain aspects of the title and key conclusions require further refinement. We have shown that 14-3-3, through its interaction with phosphorylated spastin, inhibits the degradation of spastin. Also, we have demonstrated that 14-3-3 can enhance spastin's microtubule-severing ability in cell lines. Furthermore, our work has illustrated the significant roles of 14-3-3 and spastin in the repair process of spinal cord injury. However, there is currently insufficient direct evidence to confirm the cooperation between 14-3-3 and spastin during axon regeneration and the recovery of spinal cord injury. Moreover, we have not provided conclusive evidence of their simultaneous action in injured axons, mediating changes in microtubule dynamics. Consequently, we have re-evaluated the manuscript's title and primary conclusions, and have made relevant modifications. For more detailed information, please refer to the reviewer's comments.

**Public Reviews:**

**Reviewer #1 (Public Review):**
The present work establishes 14-3-3 proteins as binding partners of spastin and suggests that this binding is positively regulated by phosphorylation of spastin. The authors show evidence that 14-3-3 - spastin binding prevents spastin ubiquitination and final proteasomal degradation, thus increasing the availability of spastin. The authors measured microtubule severing activity in cell lines and axon regeneration and outgrowth as a prompt to spastin activity. By using drugs and peptides that separately inhibit 14-3-3 binding or spastin activity, they show that both proteins are necessary for axon regeneration in cell culture and in vivo models in rats.The following is an account of the major strengths and weaknesses of the methods and results.Major strengths-The authors performed pulldown assays on spinal cord lysates using GST-spastin, then analyzed pulldowns via mass spectrometry and found 3 peptides common to various forms of 14-3-3 proteins. In co-expression experiments in cell lines, recombinant spastin co-precipitated with all 6 forms of 14-3-3 tested. The authors could also co-immunoprecipitate spastin-14-3-3 complexes from spinal cord samples and from primary neuronal cultures.-By protein truncation experiments they found that the Microtubule Binding Domain of spastin contained the binding capability to 14-3-3. This domain contained a putative phosphorylation site, and substitutions that cannot be phosphorylated cannot bind to 14-3-3.-Overexpression of GFP-spastin shows a turn-over of about 12 hours when protein synthesis is inhibited by cycloheximide. When 14-3-3 is co-overexpressed, GFP-spastin does not show a decrease by 12 hours. When S233A is expressed, a turn-over of 9 hours is observed, suggesting that phosphorylation increases the stability of the protein. In support of that notion, the phospho-mimetic S233D makes it more stable, lasting as much as the over-expression of 14-3-3.-By combining FCA with Spastazoline, authors claim that FCA increased regeneration is due to increased spastin activity in various models of neurite outgrowth and regeneration in cell culture and in vivo, the authors show impressive results on the positive effect of FCA in regeneration, and that this is abolished when spastin is inhibited.Major weaknesses1. The present manuscript suggests that 14-3-3 and spastin work in the same pathway to promote regeneration. Although the manuscript contains valuable evidence in support for a role of 14-3-3 and spasting in regeneration, the conclusive evidence is difficult to generate, and is missing in the present manuscript. For example, there are simpler explanations for the combined effect of FC-A and spastazoline. The FC-A mechanism of action can be very broad, since it will increase the binding of all 14-3-3 proteins with presumably all their substrates, hence the pathways affected can rise to the hundreds. The fact that spastazoline abolishes FC-A effect, may not be because of their direct interaction, but because spastin is a necessary component of the execution of the regeneration machinery further downstream, in line with the fact that spastazoline alone prevented outgrowth and regeneration, and in agreement with previous work showing that normal spastin activity is necessary for regeneration.With this in mind, I consider the title and most major conclusions of the manuscript related to these two proteins acting together for the observed effects are overstated.

Response: We appreciate and acknowledge the reviewers' considerations. Our results demonstrated that the spastin inhibitor, spastazolin, almost completely inhibited axon regeneration and the spinal cord injury repair process. This, in turn, leads to the disappearance of any promoting effect on spinal cord injury repair when spastin function is compromised. While we have provided evidence that the expression levels of spastin are moderately increased at the injury site in mice after treatment with FC-A following spinal cord injury, the conclusion that FC-A promotes spinal cord injury repair through the direct interaction between 14-3-3 and spastin still lacks direct evidence. Therefore, we have made appropriate modifications to the manuscript's title and main conclusions.

1. Authors show that S233D increases MT severing activity, and explain that it is related to increased binding to 14-3-3. An alternative explanation is that phosphorylation at S233 by itself could increase MT severing activity. The authors could test if purified spastin S233D alone could have more potent enzymatic activity.

Response: We appreciate the considerations of the reviewer. We believe that supplementing in vitro experiments to assess whether S233D affects spastin's microtubule severing function can more intuitively demonstrate whether phosphorylation of spastin at S233 affects its microtubule severing function; however, spastin forms hexamers through its AAA domain to exert ATPase activity and cut microtubules. Current research has reported that mutation sites leading to changes in microtubule severing function are mainly located within spastin's AAA domain (affecting spastin's ATPase activity, amino acids 342-599), such as E356A, G370R, N386K, K388R, E442Q, K427R and R562Q. Furthermore, studies have shown that mutating 11 phosphorylation sites in spastin's MIT and MTBD regions to alanine does not affect spastin's microtubule severing function, including human S268 (Rat Ser233) (Phosphorylation mutation impairs the promoting effect of spastin on neurite outgrowth without affecting its microtubule severing ability. Eur J Histochem. doi: 10.4081/ejh.2023.3594). Additionally, we also provided supplementary experiments in cell lines which showed that both spastin S233A and S233D could effectively sever microtubules (Fig.S2).

1. The interpretation of the authors cannot explain how Spastin can engage in MT severing while bound to 14-3-3 using its Microtubule Binding Domain.

Response: We appreciate the considerations of the expert reviewer. The IP experiments with truncated fragments suggest that the binding region of 14-3-3 with spastin is located within the region (215-336 amino acids) in spastin. Furthermore, experiments involving site-directed mutagenesis confirm that the actual binding site of 14-3-3 with spastin is the S233 site, rather than its MTBD region (270-328). Therefore, we have made corrections in the manuscript. We also indicate that 14-3-3 enhances spastin's protein levels by binding to the S233 site, which may be due to 14-3-3 masking the ubiquitination sites near spastin S233 (K206 or K254). Our further experiments also demonstrate that 14-3-3 inhibits the ubiquitination degradation pathway of phosphorylated spastin.

1. Also, the term "microtubule dynamics", which is present in the title and in other major conclusions, is overstated. Although authors show, in cell lines, changes in microtubule content, it is far from evidence for changes in "MT dynamics" in the settings of interest (i.e. injured axons).

Response: We appreciate and acknowledge the rigorous feedback. While our manuscript demonstrated the regulatory role of 14-3-3 and spastin in microtubule dynamics in cell lines, we lack direct evidence of these changes in microtubule dynamics within injured axons. Therefore, we have made appropriate modifications to the title, main conclusions, and related statements in our manuscript.

1. In the same lines, the manuscript lacks evidence for the changes of MT content and/dynamics as a function of the proposed 14-3-3 - Spastin pathway.

Response: We appreciate and concur with the opinions of the expert reviewer. The observed changes in microtubule dynamics in spinal cord injury were related to the overall alterations in microtubule dynamics within the spinal cord injury site. We still lack direct evidence that 14-3-3, in conjunction with spastin, alters the microtubule dynamics within axons during the process of regeneration. Therefore, we have made modifications to the manuscript.

**Reviewer #2 (Public Review):**
Summary:The idea of harnessing small molecules that may affect protein-protein interactions to promote axon regeneration is interesting and worthy of study. In this manuscript Liu et al. explore a 14-3-3-Spastin complex and its role in axon regeneration.Strengths:Some of the effects of FC-A on locomotor recovery after spinal cord contusion look interestingWeaknesses:The manuscript falls short of establishing that a 14-3-3-Spastin complex is important for any FC-A-dependent effects and there are several issues with data quality that make it difficult to interpret the results. Importantly, the effects of the spastin inhibitor has a major impact on neurite outgrowth suggesting that cells simply cannot grow in the presence of the inhibitor and raising serious questions about any selectivity for FC-A - dependent growth. Aspects of the histology following spinal cord injury were not convincing.

Response: We appreciate the rigorous review by the expert reviewers. In response to the feedback from reviewer 1, we lack direct evidence to demonstrate that the reparative effect of FC-A on spinal cord injury is mediated by the combined action of 14-3-3 and spastin. We have accordingly made the necessary changes to our manuscript. Additionally, due to upload limitations, the resolution of our tissue slices related to spinal cord injury in the manuscript is relatively low. To address this, we have supplemented relevant images which was enlarged in the supplementary materials (Fig. S7-9), Also, the original confocal files and images were uploaded.

Furthermore, our manuscript does not suggest that the reparative effect of FC-A in spinal cord injury selectively impacts the interaction between 14-3-3 and spastin. Therefore, we have modified our claims (title and conclusions) to ensure a more precise statement. Despite the fact that our axonal markers do not fully align, our evidence still strongly supports the role of FC-A in promoting nerve regeneration after spinal cord injury. Additionally, we will further optimize our immunohistochemistry methods.

**Reviewer #3 (Public Review):**
Summary:The current manuscript shows that 14-3-3 are binding partners of spastin, preventing its degradation. It is additionally shown, using complementary methods, that both 14-3-3 and spastin are necessary for axon regeneration in vitro and in vivo. While interesting in vitro and vivo data is provided, some of the claims of the authors are not convincingly supported.Major strengths:Very interesting effect of FC-A in functional recovery after spinal cord injury.Major Weaknesses:Some of the in vitro data, including colocalizations, and analysis of microtubule severing fall short to support the claims of the authors.The in vivo selectivity of FC-A towards spastin is not adequately supported by the data presented.There are aspects of the spinal cord injury site histology that are unclear.

Response: Reviewer 3's comments align with those of Reviewers 1 and 2.

**Reviewer #1 (Recommendations For The Authors):**
-The new blots presented in Fig. 3N lacks corresponding labels as for antibodies used for IP and IB and molecular weight markers.

Response: We appreciate the reviewer's feedback. We have made the corresponding modifications in the figure.

**Reviewer #2 (Recommendations For The Authors):**
The authors have addressed many of the specific concerns shared with the authors in the first round of review but several issues remain with the manuscript.1. Fig. 1D - the interpretation that spastin co-localizes with 14-3-3 proteins in hippocampal neurons is still tenuous since 14-3-3 uniformly labels the cell.

Response: We appreciate the reviewer's consideration. Upon re-examining the source files, we found that the predominant reason for 14-3-3 showing a ubiquitous cellular distribution was excessive brightness and insufficient contrast. After appropriate adjustments, we discovered that 14-3-3 exhibits characteristic distribution in axons, including aggregation at growth cone and specific locations in the axon shaft. We have made the relevant changes in the revised version.

1. Line 336. The meaning of the following statement is unclear "To further identify which isoform of 14-3-3 interacts with spastin, we generated six 14-3-3 isoforms in rats (β、γ、ε、ζ、η、θ ), then purified GST fusion 14-3-3 proteins (Figure 1G).

Response: Sorry for any confusing statement. We obtained gene fragments of six 14-3-3 isoforms from rat brain cDNA and inserted these fragments into the pEGX-5X-3 vector. Subsequently, GST 14-3-3 fusion proteins were expressed and purified in vitro. We have made the corresponding revisions in the revised version.

1. Line 341. The authors still fall short of showing that spastin and 14-3-3 interact directly thus it may be more accurate to say that they form a complex.

Response: Thank you for the reviewer's advice. We have made the corresponding corrections in the manuscript.

1. Line 388. Please clarify 2th and the meaning of "moderately" - "S233D was moderately expressed in primary hippocampal neurons at 2th DIV." While it is specified that the transfection dosage and duration were meticulously controlled - it is unclear what the criteria was for establishing the appropriate moderate dosage.

Response: Sorry about the mistake, it should be "2nd" instead of "2th". In order to establish a model for overexpressing spastin to promote neuronal neurite growth, we transfected 0.2 µg of plasmid into 1 well (1×104 cells/cm2, 24-well plate), with a transfection duration controlled at 24 hours.

1. Line 395 - It is unclear if S233D is toxic as there seem to be no measurements of cell survival.

Response: We have supplemented relevant experiments (See comment 6) based on comment 6 and found that Spastin S233D can promote neuronal neurite growth. The corresponding descriptions have been revised.

1. The pro-growth effects of S233A still does not seem to fit the narrative and the results would have been more convincing if dosage was better controlled to establish any differences between WT and S233A Spastin.

Response: We appreciate the constructive comments from the reviewer. In order to better illustrate the role of spastin S233 in neuronal growth, we have made appropriate adjustments to our experimental conditions based on previous experiments. Cells were transfected with plasmids expressing non-fused GFP and spastin and the relevant S233 mutants at a transfection dose of 0.2 µg into 1 well (1×104 cells/cm2, 24-well plate), duration was controlled at 12 hours. Due to the low expression state of the overexpressed protein, GFP (ab290 antibody for IF) was then stained to trace neuronal morphology. The experimental results demonstrate that spastin promotes neuronal neurite growth, and the dephosphorylation mutant of spastin (spastin S233A) significantly attenuates its neurite-promoting effect compared to wild-type spastin. Conversely, the phosphorylation mutant spastin S233D further enhances the promotion of neuronal neurite growth. We have also made corrections to the relevant statements in the manuscript.

1. The reason for examining protection in response to glutamate is not well rationalized based on known spastin functions. The interpretation of this experiment is unclear with respect to effects on protection vs repair.

Response: Thank you for the reviewer's consideration. We suppose that spastin may be involved in both protective and repair processes. Existing studies suggest that spastin can control store-operated calcium entry (SOCE) by altering endoplasmic reticulum morphology (doi: 10.1093/brain/awac122, doi: 10.3389/fphys.2019.01544), which may indicate its role in regulating calcium overload. Additionally, due to the critical role of spastin in axon growth, it is also essential for neuronal repair after injury. Therefore, we have not strictly distinguished between these two concepts here.

1. It is unclear if Spastazoline simply blocks any type of growth and it is thus difficult to conclude that FC-A functions through a 14-3-3-spastin effect based on the current data.

Response: We have re-evaluated and modified the title and main conclusions of the manuscript based on the reviewer's comments and the existing evidence, as responded to in reviewer 1's comments.

1. The access of FC-A to the CNS with the current protocol has not been clearly established and the effects of FC_A on spastin expression seem to mirror the profile of the control condition.

Response: We agree with the reviewer's comments. The expression trend of spastin after FC-A treatment is consistent with that of the control group, with a slight increase in its expression level compared to the control group.

1. The NF and 5-HT staining is not convincing labelling fibres.

Response: We appreciate the reviewer's comments. We believe that the reason for the incomplete axon staining is closely related to the thickness of the tissue sections. In our future research, we will further optimize our axon labeling methods.

**Reviewer #3 (Recommendations For The Authors):**
Figure 1D: Both spastin and 14-3-3 label the entire neuron which is rather unusual. Conditions of immunfluorescence should be improved. As it is, this image should not be used to claim colocalization.

Response: We appreciate the reviewer's consideration. In response to comment 1 from the expert reviewer 2, we have re-examined the source files and identified that the primary reason for the overall cell-wide distribution of 14-3-3 and spastin is due to excessive brightness and a lack of sufficient contrast. After making appropriate adjustments, we found that 14-3-3 and spastin exhibit characteristic localization within the axon (concentrated in a particular region of the axon shaft and the growth cone). We have made corresponding revisions in the revised version of the manuscript.

Figure S2: The experimental setup and data provided is not adequate to infer microtubule severing.

Response: We appreciate the reviewer's guidance. We have improved the relevant experiments and used a 100X objective lens to observe the microtubule structures more clearly.

Figure 2 I-K: The functional effect of spastin S233A and S233D on neurite outgrowth does not correlate with a function of 14-3-3 and thus does not support the central hypothesis of the manuscript. Minor: The images selected as representative show differences in neurite length and branching that are not portrayed in the graphs.

Response: Thank you for the reviewer’s comment. Similar to the response to the reviewer 2's comment 6, in order to better illustrate the role of spastin S233 in neurite outgrowth, we made corresponding adjustments to our experimental conditions. Cells were transfected with plasmids expressing non-fused GFP and spastin and the relevant S233 mutants at a transfection dose of 0.2 µg into 1 well (1×104 cells/cm2, 24-well plate), duration was controlled at 12 hours. Due to the low expression state of the overexpressed protein, GFP (ab290 antibody for IF) was then stained to trace neuronal morphology. The experimental results demonstrate that spastin promotes neuronal neurite growth, and the dephosphorylation mutant of spastin (spastin S233A) significantly attenuates its neurite-promoting effect compared to wild-type spastin. Conversely, the phosphorylation mutant spastin S233D further enhances the promotion of neuronal neurite growth. We have also made corrections to the relevant statements in the manuscript.

Figure 5 J and L: The quality, resolution and size of the images is insufficient to support the claims of the authors. As it is, one cannot interpret the data.It is very hard to envisage, even considering the explanation provided by the authors, that spinal cords where spastazoline was used correspond to contusion as a complete discontinuity between the rostral and caudal spinal cord tissue is present.

Response: Due to limitations in file uploads, we encountered issues with the resolution of the tissue slices related to spinal cord injury. To address this, we have adjusted the size and resolution of the corresponding images in the supplementary materials (Fig.S7-S9 ) and included the original confocal files and images.

Additionally, it's important to note that the tissue slices we presented do not represent all layers of the spinal cord, and not all layers exhibit discontinuity. Our slices are taken longitudinally at the dorsal site of the lesion area. The dorsal slices represent areas closer to the injury site, while deeper slices correspond to areas distant from the injury site. Therefore, we selected areas closer to the injury site to reflect the repair process following injury.

Figure 7B: Similar comment to spianl cord images provided in Figure 5. NF and MBP are not supposed to colocalize as they label different cell types...

Response: We appreciate the comments from the expert reviewer, and we agree with their suggestions. We will further optimize our axon labeling methods. The excessive brightness and lack of contrast primarily led to the non-specific labeling of other cell types with the MBP antibody. In fact, our primary goal was to highlight the injured areas by enhancing the fluorescence intensity of the images, which inadvertently resulted in neglecting the exclusion of non-specific staining. Therefore, we have made appropriate adjustments to the images to better visualize the distribution of myelin sheaths.